# Phosphorylation bar-coding of free fatty acid receptor 2 is generated in a tissue-specific manner

Natasja Barki[1], Laura Jenkins[1], Sara Marsango[1], Domonkos Dedeo[1], Daniele Bolognini[1], Louis Dwomoh[1], Aisha M Abdelmalik[1], Margaret Nilsen[1], Manon Stoffels[1], Falko Nagel[2], Stefan Schulz[2,3], Andrew B Tobin[1], Graeme Milligan[1]*

[1]Centre for Translational Pharmacology, School of Molecular Biosciences, College of Medical, Veterinary and Life Sciences, University of Glasgow, Glasgow, United Kingdom; [2]7TM Antibodies GmbH, Jena, Germany; [3]Institute of Pharmacology and Toxicology, University Hospital Jena, Jena, Germany

*For correspondence:
graeme.milligan@glasgow.ac.uk

Competing interest: The authors declare that no competing interests exist.

**Abstract** Free fatty acid receptor 2 (FFAR2) is activated by short-chain fatty acids and expressed widely, including in white adipocytes and various immune and enteroendocrine cells. Using both wild-type human FFAR2 and a designer receptor exclusively activated by designer drug (DREADD) variant we explored the activation and phosphorylation profile of the receptor, both in heterologous cell lines and in tissues from transgenic knock-in mouse lines expressing either human FFAR2 or the FFAR2-DREADD. FFAR2 phospho-site-specific antisera targeting either pSer$^{296}$/pSer$^{297}$ or pThr$^{306}$/pThr$^{310}$ provided sensitive biomarkers of both constitutive and agonist-mediated phosphorylation as well as an effective means to visualise agonist-activated receptors in situ. In white adipose tissue, phosphorylation of residues Ser$^{296}$/Ser$^{297}$ was enhanced upon agonist activation whilst Thr$^{306}$/Thr$^{310}$ did not become phosphorylated. By contrast, in immune cells from Peyer's patches Thr$^{306}$/Thr$^{310}$ become phosphorylated in a strictly agonist-dependent fashion whilst in enteroendocrine cells of the colon both Ser$^{296}$/Ser$^{297}$ and Thr$^{306}$/Thr$^{310}$ were poorly phosphorylated. The concept of phosphorylation bar-coding has centred to date on the potential for different agonists to promote distinct receptor phosphorylation patterns. Here, we demonstrate that this occurs for the same agonist-receptor pairing in different patho-physiologically relevant target tissues. This may underpin why a single G protein-coupled receptor can generate different functional outcomes in a tissue-specific manner.

## eLife assessment

In this study, the authors present **important** tools for monitoring distinct tissue-specific patterns of agonist-induced Free Fatty Acid receptor 2 phosphorylation. The work includes several validation experiments, which provide **convincing** evidence that will be beneficial for the scientific community.

## Introduction

G protein-coupled receptors (GPCRs) routinely are constitutively phosphorylated or become phosphorylated on serine and threonine residues, located either within the third intracellular loop or the C-terminal tail, following exposure to activating agonist ligands (*Zhang et al., 2022*). This is generally accompanied by new or enhanced interactions with arrestin adapter proteins. Canonically this results in receptor desensitisation because the positioning of the arrestin precludes simultaneous interactions

of the GPCR with a heterotrimeric guanine nucleotide binding (G) protein and hence 'arrests' further G protein activation (**Sun and Kim, 2021**). The earliest studies on GPCR phosphorylation determined that receptors exist in multiply phosphorylated states where numerous kinases, that include members of the G protein receptor kinase (GRK) family, second messenger-regulated kinases, and even those of the casein kinase family are involved (**Pitcher et al., 1998**; **Budd et al., 2000**; **Torrecilla et al., 2007**). Using mass spectrometry approaches and phospho-site-specific antibodies it emerged that the pattern of receptor phosphorylation was distinct among cell types and tissues (**Tran et al., 2004**; **Nobles et al., 2011**; **Kaya et al., 2020**). This led to the notion that the tissue-specific signalling output of GPCRs might be determined, at least in part, by the pattern of receptor phosphorylation – a hypothesis coined the phosphorylation barcode (**Nobles et al., 2011**; **Tobin et al., 2008**). This notion has recently been taken a step further with appreciation that the conformation adopted by an arrestin on interaction with the phosphorylated form of a GPCR might be affected by the pattern of receptor phosphorylation and in turn this may affect the signalling outputs mediated by arrestins (**Latorraca et al., 2020**). The challenge in fully appreciating the impact of the receptor phosphorylation barcode on the physiological activity of GPCRs is in the identification of the receptor-phosphorylation patterns in native tissues. Here, we address this issue by use of antibodies raised against specific phosphorylation sites within the free fatty acid receptor 2 (FFAR2).

A number of GPCRs are able to bind and respond to short chain fatty acids (SCFAs) that are generated in prodigious quantities by fermentation of dietary fibre by the gut microbiota (**Xie et al., 2023**; **Tan et al., 2023**). The most studied and best characterised of these is FFAR2 (also designated GPR43) (**Bolognini et al., 2016b**; **Stoddart et al., 2008**; **Ikeda et al., 2022**). This receptor is expressed widely, including by a range of immune cells, adipocytes, enteroendocrine, and pancreatic cells (**Bolognini et al., 2016b**; **Stoddart et al., 2008**; **Ikeda et al., 2022**). This distribution has led to studies centred on its potential role at the interface of immune cell function and metabolism (**Xie et al., 2023**; **Tan et al., 2023**; **Alvarez-Curto and Milligan, 2016**), as well as other potential roles in regulation of gut mucosal barrier permeability (**Xie et al., 2023**; **Tan et al., 2023**) and the suppression of bacterial and viral infections (**Sencio et al., 2020**; **Schlatterer et al., 2021**). In addition to being widely expressed FFAR2 is, at least when expressed in simple heterologous cell lines, able to couple to a range of different heterotrimeric G proteins from each of the $G_i$, $G_q$, and $G_{12}/G_{13}$ families (**Brown et al., 2003**; **Bolognini et al., 2019**). Despite this, however, studies suggest more selective activation of signalling pathways is induced by stimulation of FFAR2 in different native cells and tissues. For example, in white adipocytes stimulation of FFAR2 is clearly anti-lipolytic, an effect mediated via pertussis toxin-sensitive $G_i$ proteins and reduction in cellular cAMP levels (**Bolognini et al., 2016a**), whilst in GLP-1-positive enteroendocrine cells activation of FFAR2 promotes release of this incretin hormone in a $Ca^{2+}$ and $G_q$-mediated manner (**Bolognini et al., 2016a**).

A challenge in studying the molecular basis of effects of SCFAs in native cells and tissues is that the pharmacology of the various GPCRs that respond to these ligands is very limited (**Milligan et al., 2017a**; **Milligan et al., 2017b**) and there is a particular dearth of antagonist ligands for anything other than the human ortholog of FFAR2 (**Milligan et al., 2017a**; **Milligan et al., 2017b**). Designer receptor exclusively activated by designer drug (DREADD) forms of GPCRs contain mutations that eliminate binding and responsiveness to endogenously generated orthosteric agonists whilst, in parallel, allowing binding and activation by defined, but non-endogenously produced, ligands that do not activate the wild-type form of the receptor (**Wang et al., 2021**; **Bradley et al., 2018**; **Shchepinova et al., 2020**). Such DREADDs have become important tools to probe receptor function in the context of cell and tissue physiology. Here, in addition to studying wild-type human FFAR2 we also employed a DREADD form of human FFAR2 (hFFAR2-DREADD) that we have previously generated (**Hudson et al., 2012a**) and characterised extensively (**Bolognini et al., 2019**; **Barki et al., 2022**; **Milligan et al., 2021**). This has the benefit of providing both small, highly selective, water-soluble orthosteric agonists for FFAR2 and, when using the human ortholog of FFAR2, there are high affinity and well-characterised antagonists (**Milligan et al., 2017a**; **Milligan et al., 2017b**; **Sergeev et al., 2017**) that can be used to further define on-target specificity of effects that are observed.

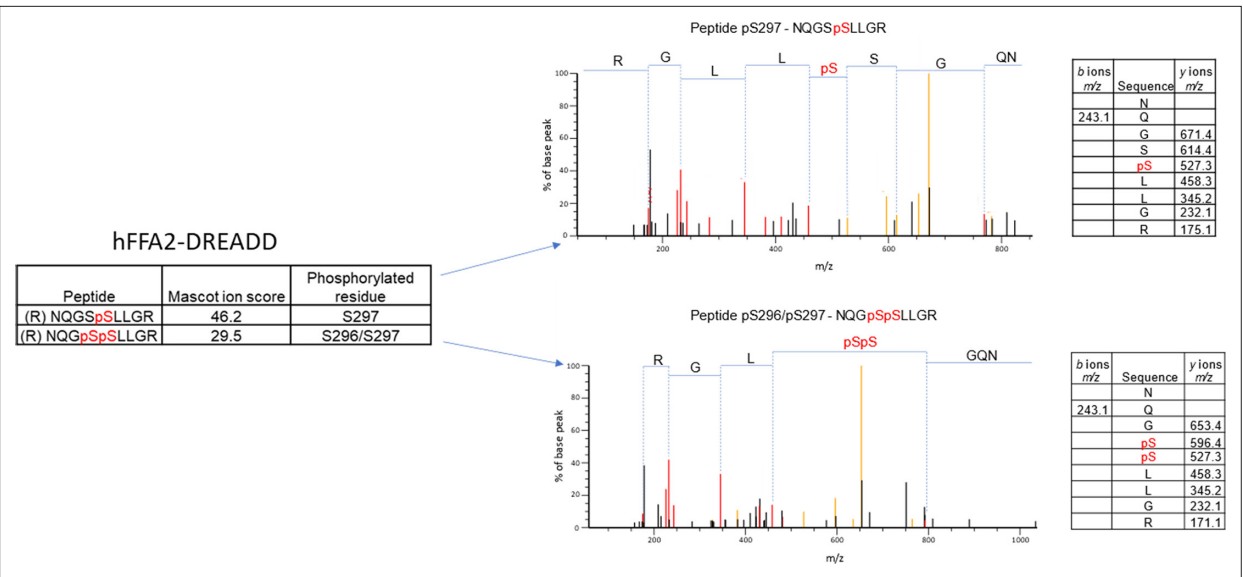

**Figure 1.** Mass spectrometry analysis of hFFAR2-DREADD-eYFP identifies basal phosphorylation of Ser[297] and agonist-promoted phosphorylation of Ser[296]. Mass spectrometry analysis was conducted on samples isolated from Flp-In T-REx 293 cells in which expression of hFFAR2-DREADD-eYFP had been induced. Experiments were performed on vehicle and 4-methoxy-3-methyl-benzoic acid (MOMBA)-treated (100 μM, 5 min) cells as detailed in Materials and methods. LC-MS/MS identified Ser[297] as being phosphorylated constitutively, and Ser[296/297] as being phosphorylated by sorbic acid or MOMBA. Composite outcomes of a series of independent experiments are combined. Fragmentation tables associated with phosphorylated peptides are shown. Phosphorylated residues are highlighted in *red*.

## Results

### Generation and characterisation of phospho-site-specific antisera to identify activated hFFAR2-DREADD

We performed mass spectrometry on a form of hFFAR2-DREADD with C-terminally linked enhanced yellow fluorescent protein (hFFAR2-DREADD-eYFP) following its doxycycline-induced expression in Flp-In T-REx 293 cells. These studies identified phosphorylation at residue Ser[297] in both the basal state and after addition of the orthosteric hFFAR2-DREADD agonists (2E,4E)-hexa-2,4-dienoic acid (sorbic acid) (*Bolognini et al., 2019*; *Hudson et al., 2012a*) or 4-methoxy-3-methyl-benzoic acid (MOMBA) (*Barki et al., 2022*). In addition, the adjacent residue Ser[296] was also observed to be phosphorylated only after agonist treatment (*Figure 1*). Based on these outcomes we generated an antiserum anticipated to identify hFFAR2-DREADD when either pSer[296], pSer[297], or both these amino acids, were phosphorylated (*Figure 2A and B*). Ser[296] and Ser[297] are within the intracellular C-terminal tail of the receptor (*Figure 2A*) and within this region there are other potential phospho-acceptor sites. These include Thr[306] and Thr[310] as well as Ser[324] and Ser[325] (*Figure 2A*). Although we did not obtain clear evidence of either basal or DREADD agonist-induced phosphorylation of these residues from the mass spectrometry studies, we also generated antisera potentially able to identify either pThr[306]/pThr[310] hFFAR2-DREADD (*Figure 2A and B*) or pSer[324]/pSer[325] hFFAR2-DREADD (not shown).

To initially assess these antisera we performed immunoblots using membrane preparations from the same Flp-In T-REx 293 cells that had been induced to express hFFAR2-DREADD-eYFP. The potential pSer[296]/pSer[297] antiserum was able to identify a diffuse set of polypeptides centred at 70 kDa in preparations generated from vehicle-treated cells (*Figure 2B*) and the intensity of staining was increased in samples derived from cells after exposure to the hFFAR2-DREADD agonist MOMBA (*Figure 2B*). By contrast when samples were produced from a Flp-In T-REx 293 cell line induced to express a variant of hFFAR2-DREADD-eYFP in which both Ser[296] and Ser[297] were converted to Ala (hFFAR2-DREADD-PD1) (*Figure 2A and B*) the potential pSer[296]/pSer[297] antiserum was unable to identify the receptor protein either with or without exposure of the cells to MOMBA (*Figure 2B*). Similar studies were performed with the putative pThr[306]/pThr[310] antiserum on samples induced to express hFFAR2-DREADD-eYFP. In this case there was no significant detection of the receptor construct in samples from vehicle-treated cells (*Figure 2B*), however, after exposure to MOMBA there was also strong identification of diffuse

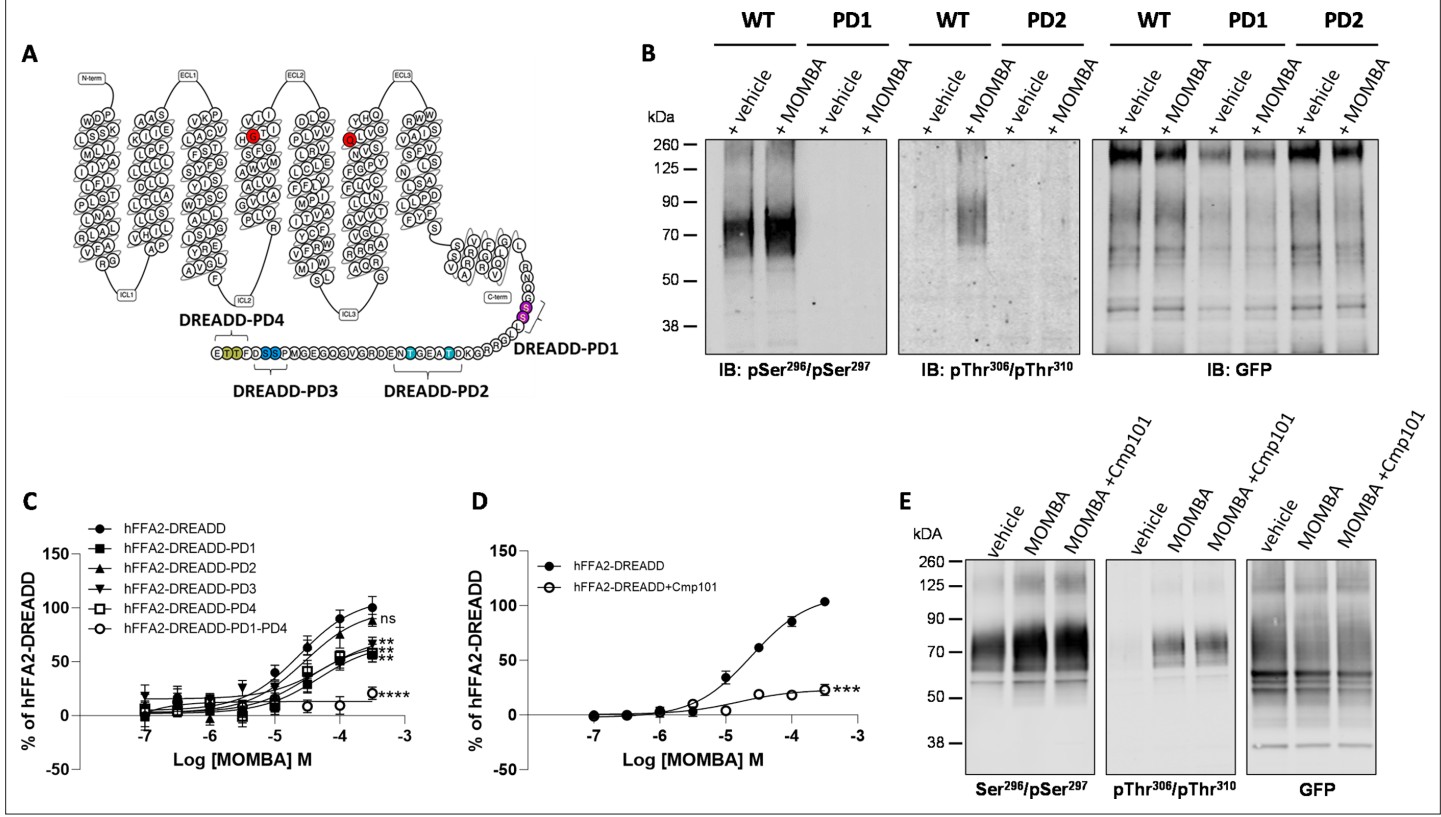

**Figure 2.** Characteristics of putative pSer$^{296}$/pSer$^{297}$ and pThr$^{306}$/pThr$^{310}$ hFFAR2-antisera and the effect of potential phospho-acceptor site mutations on agonist-induced arrestin-3 interactions. The primary amino acid sequence of hFFAR2 is shown (**A**). Residues altered to generate the DREADD variant are in red (Cys$^{141}$Gly, His$^{242}$Gln). Phospho-deficient (PD) hFFAR2-DREADD variants were generated by replacing serine 296 and serine 297 (purple, hFFAR2-DREADD-PD1), threonine 306 and threonine 310 (light blue, hFFAR2-DREADD-PD2), serine 324 and serine 325 (dark blue, hFFAR2-DREADD-PD3) or threonine 328 and 329 (yellow, hFFAR2-DREADD-PD4) with alanine. In addition, hFFAR2-DREADD-PD1-4 was generated by combining all these alterations. (**B**) The ability of putative pSer$^{296}$/pSer$^{297}$ and pThr$^{306}$/Thr$^{310}$ antisera to identify wild-type and either PD1 or PD2 forms of hFFAR2-DREADD with and without treatment of cells expressing the various forms with 4-methoxy-3-methyl-benzoic acid (MOMBA) is shown and, as a control, anti-GFP immunoblotting of equivalent samples is illustrated. (**C**) The ability of varying concentrations of MOMBA to promote interaction of arrestin-3 with hFFAR2-DREADD and each of the DREADD-PD mutants is illustrated. Each of the DREADD-PD variants, except hFFAR2-DREADD-PD2 (ns), were less effective in promoting interactions in response to MOMBA (**$p<0.01$, ****$p<0.0001$). (**D**) The effect of the GRK2/3 inhibitor compound 101 on the capacity of MOMBA to promote recruitment of arrestin-3 to wild-type hFFAR2-DREADD is shown (***$p<0.001$). Significance in **C** and **D** were assessed by one-way ANOVA followed by Dunnett's multiple comparisons test. (**E**) The effect of compound 101 on detection of hFFAR2-DREADD-eYFP by each of pSer$^{296}$/pSer$^{297}$, pThr$^{306}$/Thr$^{310}$, and anti-GFP antisera is shown. Data are representative (**B, E**) or show means ± SEM (**C, D**) of at least three independent experiments.

The online version of this article includes the following source data for figure 2:

**Source data 1.** The ability of putative pSer$^{296}$/pSer$^{297}$ antisera to identify wild-type and PD1 forms of hFFAR2-DREADD.

**Source data 2.** The ability of putative and pThr$^{306}$/Thr$^{310}$ antisera to identify wild-type and PD2 forms of hFFAR2-DREADD.

**Source data 3.** GFP control for phosphor-antisera to identify wild-type and mutant forms of hFFAR2-DREADD.

**Source data 4.** 4-Methoxy-3-methyl-benzoic acid (MOMBA) promotes interaction of arrestin-3 with hFFAR2-DREADD and each of the DREADD-PD mutants.

**Source data 5.** The effect of compound 101 on detection of hFFAR2-DREADD-eYFP by pSer$^{296}$/pSer$^{297}$.

**Source data 6.** The effect of compound 101 on detection of hFFAR2-DREADD-eYFP by pThr$^{306}$/Thr$^{310}$.

**Source data 7.** GFP control for phosphor-antisera to identify the effect of compound 101.

polypeptide(s) centred at 70 kDa corresponding to hFFAR2-DREADD-eYFP (*Figure 2B*). Such staining was absent, however, when a variant hFFAR2-DREADD-eYFP, in which in this case both Thr$^{306}$ and Thr$^{310}$ were altered to Ala (hFFAR2-DREADD-PD2), was induced (*Figure 2A and B*). To ensure that the lack of recognition by the potential phospho-site-specific antisera was not simply due to poor expression of either the hFFAR2-DREADD-PD1 or hFFAR2-DREADD-PD2 variants, we also immunoblotted

equivalent samples with an anti-GFP antiserum (that also identifies eYFP). This showed similar levels of detection of the approximately 70 kDa polypeptide(s) in all samples (*Figure 2B*). Although as noted earlier we did attempt to generate a potential pSer$^{324}$/pSer$^{325}$ antiserum, we were unable to detect the receptor using this in samples containing hFFAR2-DREADD-eYFP exposed to either MOMBA or vehicle (data not shown). As such, this was not explored further. However, as we also generated both a hFFAR2-DREADD-PD3 variant, in which both Ser$^{324}$ and Ser$^{325}$ were altered to alanine (*Figure 2A*), and an hFFAR2-DREADD-PD4 variant (*Figure 2A*) in which both Thr$^{328}$ and Thr$^{329}$ were altered to alanine, we assessed the effects of these changes on the capacity of MOMBA to promote interactions of wild-type and the phospho-deficient (PD) variants of hFFAR2-DREADD-eYFP with arrestin-3. MOMBA promoted such interactions with the intact DREADD receptor construct in a concentration-dependent manner with pEC$_{50}$=4.62±0.13 M (mean± SEM, n=3). The maximal effect, but not measured potency, of MOMBA was reduced for all the PD variants except hFFAR2-DREADD-PD2 (*Figure 2C*), but in no case was the effect on maximal signal reduced by more than 50%. We hence combined these PD variants to produce a form of the receptor in which all potential phospho-acceptor sites in the C-terminal tail were altered to alanines (hFFAR2-DREADD-PD1-4). This variant was poorly able to recruit arrestin-3 (20.8±2.3% of wild-type receptor, mean ± SD, n=3) in response to addition of MOMBA (*Figure 2C*). It is likely that GRK2 and/or GRK3 played a key role in allowing MOMBA-induced interaction with arrestin-3 because this was greatly reduced when cells expressing the wild-type form of the receptor construct were pre-treated with the GRK2/GRK3 selective inhibitor compound 101 (*Marsango et al., 2022*; *Figure 2D*). To extend this analysis we pre-treated cells induced to express hFFAR2-DREADD-eYFP with either vehicle or compound 101 and then, after addition of MOMBA, assessed receptor phosphorylation as detected by the putative pThr$^{306}$/pThr$^{310}$ and pSer$^{296}$/pSer$^{297}$ antisera. Pre-treatment with compound 101 did not substantially reduce MOMBA-mediated identification by either the pThr$^{306}$/pThr$^{310}$ antiserum or the pSer$^{296}$/pSer$^{297}$ antiserum (*Figure 2E*). This suggests that phosphorylation of these sites are not controlled by GRK2/3 and that these are not, at least in isolation, sufficient to define interactions of the receptor with arrestin-3.

## Effects of MOMBA on antisera recognition are both on-target and reflect receptor phosphorylation

The selective effect of the hFFAR2-DREADD agonist MOMBA in promoting enhanced phosphorylation of Ser$^{296}$/Ser$^{297}$ and in allowing phosphorylation of Thr$^{306}$/Thr$^{310}$ was further substantiated because both these effects of MOMBA were re-capitulated by the second hFFAR2-DREADD-specific agonist, sorbic acid (*Bolognini et al., 2019*; *Hudson et al., 2012a*; *Figure 3A*). By contrast, propionic acid (C3) which, as an SCFA, is an endogenous activator of wild-type hFFAR2 but does not activate hFFAR2-DREADD (*Bolognini et al., 2019*), was unable to modulate the basal levels of phosphorylation detected by the pSer$^{296}$/pSer$^{297}$ antiserum or to promote detection of the receptor by the pThr$^{306}$/pThr$^{310}$ antiserum (*Figure 3A*). Use of the anti-GFP antiserum confirmed similar levels of receptor in the vehicle and C3-treated samples as in those treated with MOMBA or sorbic acid and, in addition, the lack of expression of the receptor construct prior to doxycycline induction (*Figure 3A*).

To further confirm that the effects of MOMBA were due to direct activation of hFFAR2-DREADD-eYFP, we assessed whether the hFFAR2 receptor orthosteric antagonist/inverse agonist CATPB ((*S*)-3-(2-(3-chlorophenyl)acetamido)-4-(4-(trifluoromethyl)phenyl)butanoic acid) (*Hudson et al., 2012b*; *Hudson et al., 2013*) would be able to prevent the effects of MOMBA. Indeed, pre-addition of CATPB (10 mM) to cells induced to express hFFAR2-DREADD-eYFP prevented the effects of MOMBA on receptor recognition by both the putative pSer$^{296}$/pSer$^{297}$ and the pThr$^{306}$/pThr$^{310}$ antisera (*Figure 3B*). Additionally, pre-addition of CATPB lowered the extent of basal detection of the receptor by the pSer$^{296}$/pSer$^{297}$ antiserum (*Figure 3B*). As CATPB possesses inverse agonist properties (*Hudson et al., 2012b*) this may indicate that at least part of the basal pSer$^{296}$/pSer$^{297}$ signal reflects agonist-independent, constitutive activity of hFFAR2-DREADD-eYFP. Once more, parallel immunoblots using anti-GFP confirmed that there were similar levels of the receptor construct in vehicle and MOMBA±CATPB-treated samples (*Figure 3B*). To conclude the initial characterisation of the antisera we confirmed that both the putative pSer$^{296}$/pSer$^{297}$ and pThr$^{306}$/pThr$^{310}$ antisera were detecting phosphorylated states of hFFAR2-DREADD-eYFP. To do so, after exposure of cells induced to express hFFAR2-DREADD-eYFP to either vehicle or MOMBA, samples were treated with LPP prior to SDS-PAGE and immunoblotting. This treatment is anticipated to remove phosphate from proteins in a

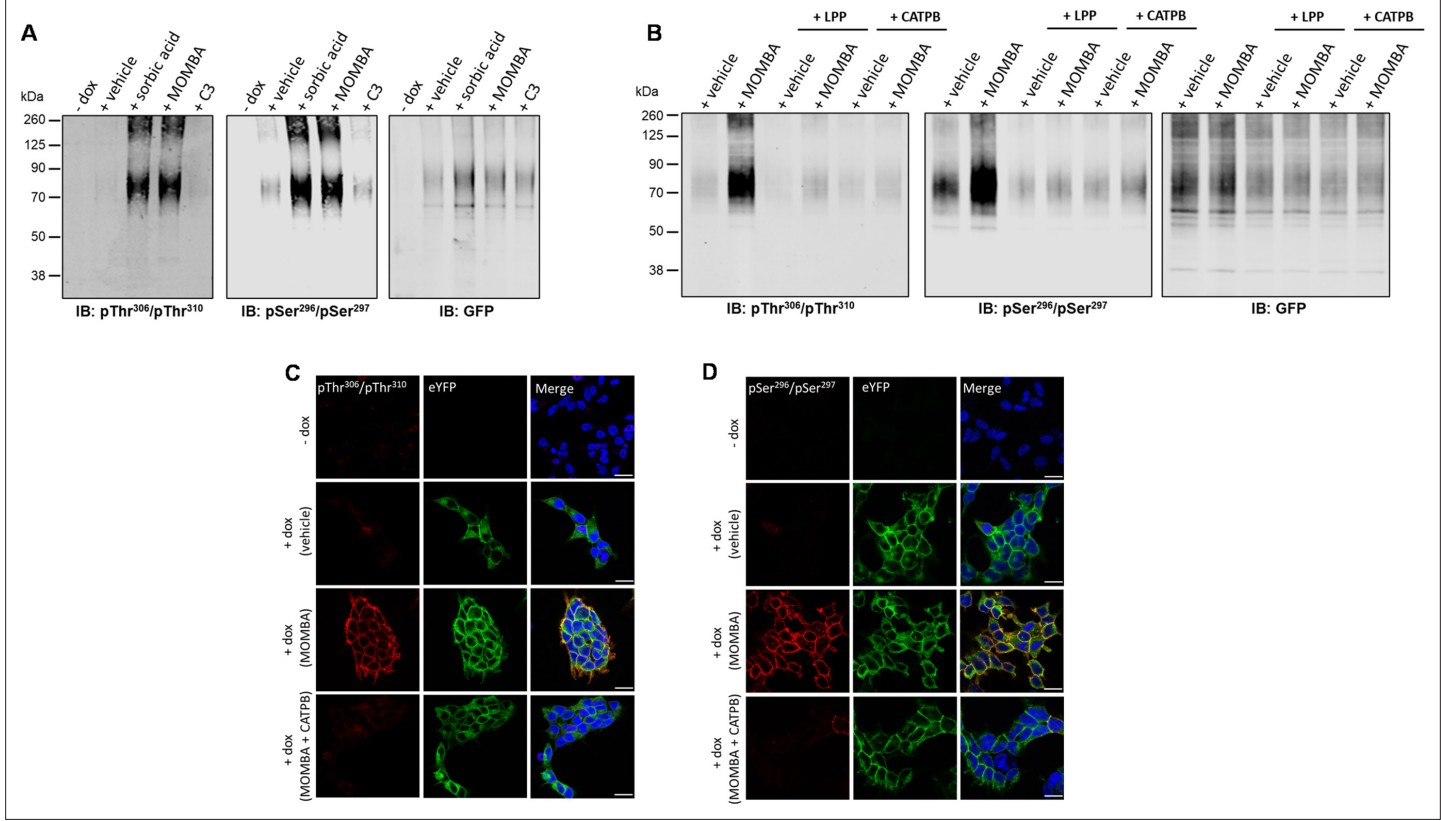

**Figure 3.** Agonist-induced detection of hFFAR2-DREADD with putative pSer$^{296}$/pSer$^{297}$ and pThr$^{306}$/pThr$^{310}$ antisera reflects receptor activation, receptor phosphorylation, and can be detected in situ. The ability of the pSer$^{296}$/pSer$^{297}$, pThr$^{306}$/pThr$^{310}$ hFFAR2, and, as a control GFP, antisera to identify hFFAR2-DREADD-eYFP after induction to express the receptor construct and then treatment of cells with vehicle, 4-methoxy-3-methyl-benzoic acid (MOMBA), sorbic acid (each 100 µM), or propionate (C3) (2 mM) is shown. In the '-dox' lanes receptor expression was not induced. (**B**) As in (**A**) except that after cell treatment with vehicle or MOMBA, immune-enriched samples were treated with lambda protein phosphatase (LPP) or, rather than treatment with MOMBA, cells were treated with a combination of MOMBA and the hFFAR2 inverse agonist ((S)-3-(2-3-chlorophenyl)acetamido)-4-(4-(trifluoromethyl)phenyl)butanoic acid (CATPB) (10 µM, 30 min pre-treatment). (**C, D**) Cells harboring hFFAR2-DREADD-eYFP and grown on glass coverslips were either untreated (- **dox**) or induced to express hFFAR2-DREADD-eYFP. The induced cell samples were then exposed to vehicle, MOMBA (100 µM), or a combination of MOMBA (100 µM) and CATPB (10 µM) for 5 min. Fixed cells were then treated with anti-pThr$^{306}$/pThr$^{310}$ (**C**) or anti-pSer$^{296}$/pSer$^{297}$ (**D**) (red, Alexa Fluor 647) or imaged to detect eYFP (green). DAPI was added to detect DNA and highlight cell nuclei (blue). Scale bar = 20 µm.

The online version of this article includes the following source data for figure 3:

**Source data 1.** Agonist-induced detection of hFFAR2-DREADD with pTHr$^{306}$/pThr$^{310}$ antisera.

**Source data 2.** Agonist-induced detection of hFFAR2-DREADD with pSer$^{296}$/pSer$^{297}$ antisera.

**Source data 3.** GFP control for agonist-induced detection of hFFAR2-DREADD.

**Source data 4.** Activation and inhibition of hFFAR2-DREADD with pThr$^{306}$/pThr$^{310}$ antisera.

**Source data 5.** Activation and inhibition of hFFAR2-DREADD with pSer$^{296}$/pSer$^{297}$ antisera.

**Source data 6.** GFP control for activation and inhibition of hFFAR2-DREADD.

site-agnostic manner. Now, MOMBA-induced detection of the receptor by both pSer$^{296}$/pSer$^{297}$ and pThr$^{306}$/pThr$^{310}$ antisera was eliminated (**Figure 3B**), as was most of the agonist-independent detection by pSer$^{296}$/pSer$^{297}$ (**Figure 3B**).

## MOMBA-induced phosphorylation of hFFAR2-DREADD-eYFP is also detected in immunocytochemical studies

The pThr$^{306}$/pThr$^{310}$ antiserum was also effective in detecting MOMBA-induced post-activation states of hFFAR2-DREADD-eYFP in immunocytochemistry studies. When induced in Flp-In T-REx 293 cells the presence of the receptor construct could be observed via the eYFP tag with or without exposure to MOMBA (**Figure 3C**). However, also in this setting, the anti-pThr$^{306}$/pThr$^{310}$ antiserum was only

able to identify the receptor after treatment with MOMBA (*Figure 3C*) and, once again, pre-addition of CATPB prevented the effect of MOMBA (*Figure 3C*) without affecting direct identification and imaging of the receptor via the eYFP tag. Similar studies were performed with the hFFAR2 pSer[296]/pSer[297] antiserum with very similar outcomes. In this setting identification of hFFAR2 DREADD-eYFP by the pSer[296]/pSer[297] antiserum was almost entirely dependent on exposure to MOMBA (*Figure 3D*) and this was also prevented by pre-treatment with CATPB (*Figure 3D*).

## Phospho-specific antisera differentially identify hFFAR2-DREADD in mouse tissues

To assess physiological roles of FFAR2 in mouse tissues without potential confounding effects of either activation of the related SCFA receptor FFA3 or non-receptor-mediated effects of SCFAs, we recently developed an hFFAR2-DREADD knock-in transgenic mouse line (*Milligan et al., 2021*). Here, mouse FFAR2 is replaced by hFFAR2-DREADD with, in addition, an appended C-terminal anti-HA epitope tag sequence to allow effective identification of cells expressing the receptor construct (*Bolognini et al., 2019*; *Barki et al., 2022*).

### White adipose tissue

FFAR2 is known to be expressed in white adipose tissue and we previously used these hFFAR2-DREADD-HA mice to define the role of this receptor as an anti-lipolytic regulator (*Bolognini et al.,*

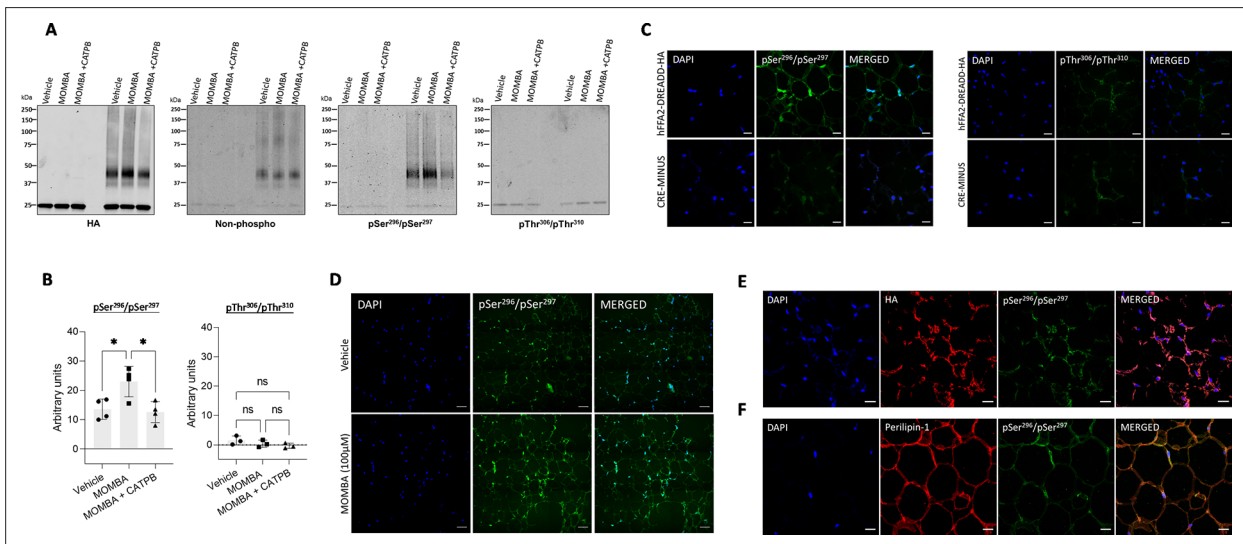

**Figure 4.** In white adipose tissue residues Ser[296]/Ser[297] of hFFAR2-DREADD-HA but not Thr[306]/Thr[310] become phosphorylated in response to 4-methoxy-3-methyl-benzoic acid (MOMBA). White adipose tissue dissected from hFFAR2-DREADD-HA and CRE-MINUS mice was treated with either vehicle, 100 μM MOMBA, or 100 μM MOMBA+10 μM ((*S*)-3-(2-3-chlorophenyl)acetamido)-4-(4-(trifluoromethyl)phenyl)butanoic acid (CATPB). (**A**) Lysates were prepared and solubilised. hFFAR2-DREADD-HA was immunoprecipitated using an anti-HA monoclonal antibody and following SDS-PAGE immunoblotted to detect HA, non-phosphorylated hFFAR2-DREADD-HA, and pSer[296]/pSer[297] or pThr[306]/pThr[310]hFFAR2-DREADD-HA. A representative experiment is shown. (**B**) Quantification of pSer[296]/pSer[297] (left) and pThr[306]/pThr[310] immunoblots (right) phosphorylation (means ± SEM) in experiments using tissue from different mice, *p<0.05, ns: not significant. Significance was assessed by one-way ANOVA, followed by Tukey's multiple comparisons test (n=4). (**C**) Tissue samples from hFFAR2-DREADD-HA (top panel) and CRE-MINUS (bottom panel) mice that were treated with MOMBA were immunostained to detect pSer[296]/pSer[297] (left panels), pThr[306]/pThr[310] (right panels) and counterstained with DAPI (blue). Scale bars = 20 μm. (**D**) Comparison of pSer[296]/pSer[297] staining of samples from hFFAR2-DREADD-HA-expressing mice vehicle treated (top panels) or treated with MOMBA (bottom panels) (scale bar = 50 μm). (**E, F**) Tissue sections from hFFAR2-DREADD-HA-expressing mice immunostained with pSer[296]/pSer[297] (green) and anti-HA (red) (**E**) to detect the receptor expression or **F** with anti-perilipin-1 (red) to identify adipocytes. Merged images are shown to the right. Scale bars = 20 μm.

The online version of this article includes the following source data for figure 4:

**Source data 1.** hFFAR2-DREADD-HA expression in white adipose tissue.

**Source data 2.** pSer[296]/pSer[297] detects phosphorylation of hFFAR2-DREADD-HA in white adipose tissue.

**Source data 3.** pThr[306]/pThr[310] is not phosphorylated in white adipose tissue.

**Source data 4.** Quantification of pSer[296]/pSer[297] and pThr[306]/pThr[310] immunoblots in white adipose tissue.

*2019*). To explore the expression and regulation of hFFAR2-DREADD-HA more fully, we took advantage of the appended HA tag to immunoprecipitate the receptor from white adipose tissue taken from hFFAR2-DREADD-HA mice. As a control, equivalent pull-down studies were performed with tissue from 'CRE-MINUS' animals (*Bolognini et al., 2019*; *Barki et al., 2022*). These harbor hFFAR2-DREADD-HA at the same genetic locus but expression has not been induced and hence they lack protein corresponding to either hFFAR2-DREADD-HA or mouse FFAR2 (*Bolognini et al., 2019*). In all of these studies, and those using other mouse-derived tissues (see later), as well as an anti-protease cocktail, we included the phosphatase inhibitor cocktail PhosSTOP, as described previously (*Fritzwanker et al., 2023*), to prevent potential dephosphorylation of hFFAR2-DREADD-HA and other proteins. Following SDS-PAGE of such immune precipitates from hFFAR2-DREADD-HA mice immunoblotting with an HA antibody resulted in detection of hFFAR2-DREADD-HA predominantly as an approximately 45 kDa species, with lower levels of a 40 kDa form (*Figure 4A*). These both clearly corresponded to forms of hFFAR2-DREADD-HA as such immunoreactive polypeptides were lacking in pull-downs from white adipose tissue from CRE-MINUS mice (*Figure 4A*). Moreover, parallel immunoblotting with an antiserum against the non-post-translationally modified C-terminal tail sequence of human FFAR2 identified the same polypeptides as the HA antibody and, once more, this was only in tissue from the hFFAR2-DREADD-HA and not CRE-MINUS animals (*Figure 4A*). Notably, immunoblotting of such samples with the anti-pSer$^{296}$/pSer$^{297}$ antiserum also detected the same polypeptides. However, unlike the hFFAR2 C-terminal tail antiserum, although the pSer$^{296}$/pSer$^{297}$ antiserum did identify hFFAR2-DREADD-HA without prior addition of a ligand, pre-addition of MOMBA increased immune detection of the receptor (*Figure 4A*). This is consistent with the ligand promoting quantitatively greater levels of phosphorylation of these residues (*Figure 4B*). Moreover, pre-addition of the hFFAR2 antagonist/inverse agonist CATPB not only prevented the MOMBA-induced enhanced detection of the receptor protein by the pSer$^{296}$/pSer$^{297}$ antiserum but showed a trend to reduce this below basal levels (*Figure 4A*) although this did not reach statistical significance (*Figure 4B*). In contrast, the anti-pThr$^{306}$/pThr$^{310}$ antiserum failed to detect immunoprecipitated hFFAR2-DREADD-HA from white adipose tissue in either the basal state or post-addition of MOMBA (*Figure 4A and B*). This indicates that these sites are not and do not become phosphorylated in this tissue.

Immunohistochemical studies performed on fixed adipose tissue from hFFAR2-DREADD-HA-expressing mice that had been pre-exposed to MOMBA were consistent with the immunoblotting studies. Clear identification of pSer$^{296}$/pSer$^{297}$ hFFAR2-DREADD-HA was detected and the specificity of this was confirmed by the absence of staining in equivalent samples from CRE-MINUS mice (*Figure 4C*). Once more, no specific staining was observed for the anti-pThr$^{306}$/pThr$^{310}$ antiserum (*Figure 4C*). Clear detection of pSer$^{296}$/pSer$^{297}$ immunostaining was observed in adipose tissue from hFFAR2-DREADD-HA mice without exposure to MOMBA but once more the intensity of staining was markedly increased after MOMBA treatment (*Figure 4D*). Parallel anti-HA staining showed strong co-localisation with the pSer$^{296}$/pSer$^{297}$ antiserum (*Figure 4E*), and clear co-localisation of anti- pSer$^{296}$/pSer$^{297}$ immunostaining with that for perilipin-1 (*Figure 4F*) confirmed the presence of pSer$^{296}$/pSer$^{297}$ hFFAR2-DREADD-HA directly on adipocytes.

## Peyer's patch immune cells

As the apparent lack of anti-pThr$^{306}$/pThr$^{310}$ immunostaining after treatment with MOMBA in white adipocytes was distinct from the outcomes observed in Flp-In T-REx 293 cells, we turned to a second source of tissue in which FFAR2 expression is abundant. Peyer's patches act as immune sensors of the gut (*Park et al., 2023*). Anti-HA staining showed extensive and high-level expression of hFFAR2-DREADD-HA in cells within these structures (*Figure 5A*). Co-staining for CD11c indicated many of the hFFAR2-DREADD-HA-positive cells correspond to dendritic cells, monocytes, and/or macrophages (*Figure 5B*). Co-staining to detect the presence of the nuclear transcription factor RoRγt also indicated, as shown previously at the mRNA level for mouse FFAR2 (*Chun et al., 2019*), that hFFAR2-DREADD-HA is well expressed by type-III innate lymphoid cells (*Figure 5C*). HA pull-down immune-captured hFFAR2-DREADD-HA from Peyer's patches and associated mesenteric lymph nodes of hFFAR2-DREADD-HA-expressing mice (*Figure 5D*). Anti-HA immunoblotting of such material indicated a more diffuse pattern of immunostaining than observed from white adipose tissue (*Figure 5D*). This may reflect a more complex pattern of post-translational modifications, including differing extents of N-glycosylation (see later) than observed in white adipose tissue (compare *Figure 5D* with

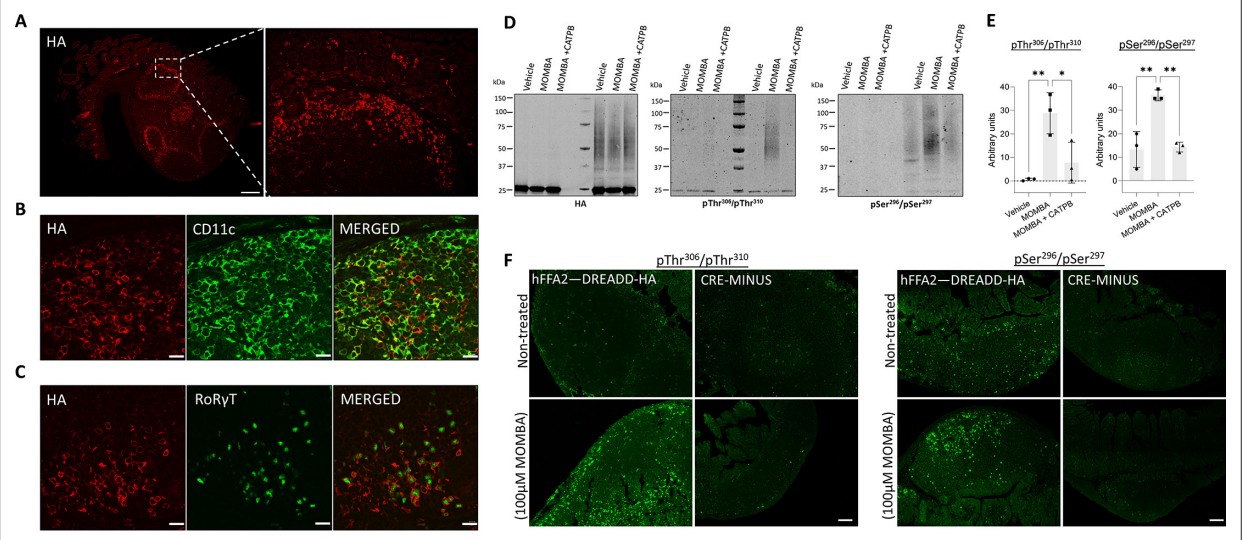

**Figure 5.** hFFAR2-DREADD-HA becomes phosphorylated at Thr[306]/Thr[310] in addition to Ser[296]/Ser[297] in immune cells from Peyer's patches. Peyer's patches isolated from hFFAR2-DREADD-HA-expressing mice were immunostained with anti-HA (red) to detect receptor expression. Images were acquired with ×20 (left panel) and ×63 (right panel) objectives (scale bar = 200 μm) (**A**). Tissue sections were counterstained with (**B**) anti-CD11c as a marker of dendritic cells, monocytes, and/or macrophages or (**C**) RORγT to detect type-III innate lymphoid cells (scale bar = 20 μm). Isolated Peyer's patches and mesenteric lymph nodes from CRE-MINUS and hFFAR2-DREADD-HA mice were exposed to either vehicle, 100 μM 4-methoxy-3-methyl-benzoic acid (MOMBA) or 100 μM MOMBA+10 μM ((*S*)-3-(2-3-chlorophenyl)acetamido)-4-(4-(trifluoromethyl)phenyl)butanoic acid (CATPB). (**D**) Following lysate preparation, immunoprecipitation and SDS-PAGE samples were probed to detect HA, pThr[306]/pThr[310], or pSer[296]/pSer[297]. (**E**) Quantification of pThr[306]/pThr[310] (left) and pSer[296]/pSer[297] immunoblots (right) hFFAR2-DREADD-HA (means ± SEM), *p<0.05, **p<0.01. Significance was assessed by one-way ANOVA, followed by Tukey's multiple comparisons test (n=3). (**F**) Treated tissue sections were also used in immunohistochemical studies, employing either pThr[306]/pThr[310] (left panels) or pSer[296]/pSer[297] (right panels) (scale bars = 100 μm).

The online version of this article includes the following source data and figure supplement(s) for figure 5:

**Source data 1.** hFFAR2-DREADD-HA expression in Peyer's patches.

**Source data 2.** pThr[306]/pThr[310] detects agonist-dependent phosphorylation in Peyer's patches.

**Source data 3.** pSer[296]/pSer[297] detects both constitutive and agonist-dependent phosphorylation in Peyer's patches.

**Source data 4.** Quantification of pSer[296]/pSer[297] and pThr[306]/pThr[310] immunoblots in Peyer's patches.

**Figure supplement 1.** hFFAR2-DREADD-HA becomes phosphorylated at Thr[306]/Thr[310] in immune cells within Peyer's patches.

*Figure 4A*). Once more, however, this range of HA-detected polypeptides did indeed all represent forms of hFFAR2-DREADD-HA as they were completely absent from HA immunocapture conducted in equivalent tissue from the CRE-MINUS animals (*Figure 5D*). Parallel immunoblots of such samples with the anti-pThr[306]/pThr[310] antiserum now indicated that, in contrast to adipocytes, hFFAR2-DREADD-HA became phosphorylated on these residues in a MOMBA-dependent manner, with no detection of phosphorylation of these residues without agonist treatment (*Figure 5D*). Moreover, this effect of MOMBA was clearly mediated by the hFFAR2-DREADD-HA receptor because anti-pThr[306]/pThr[310] recognition was entirely lacking when MOMBA was added after the addition of the hFFAR2 antagonist/inverse agonist CATPB (*Figure 5D and E*) that blocks this receptor with high affinity (*Hudson et al., 2012b*; *Hudson et al., 2013*). Similar outcomes were obtained in this tissue with the anti-pSer[296]/pSer[297] antiserum. Basal detection of the immune-captured receptor was low and this was increased markedly by pre-treatment with MOMBA (*Figure 5D and E*). As for the anti-pThr[306]/pThr[310] antiserum the effect of MOMBA on recognition of the receptor was not observed when cells were exposed to CATPB in addition to MOMBA (*Figure 5D and E*). Immunostaining of fixed Peyer's patch tissue with the anti-pThr[306]/pThr[310] antiserum confirmed the marked agonist dependence of recognition of hFFAR2-DREADD-HA in such cells (*Figure 5F*). We detected a low level of immunostaining with this antiserum in Peyer's patch tissue in the absence of addition of MOMBA but, as we detected a similar level of staining in tissue from CRE-MINUS animals, in both the presence and absence of MOMBA, this may represent a small level of non-specific reactivity (*Figure 5F*). In this tissue, immunostaining with the pSer[296]/pSer[297] antiserum indicated a degree of agonist-independent phosphorylation of these sites as this was not

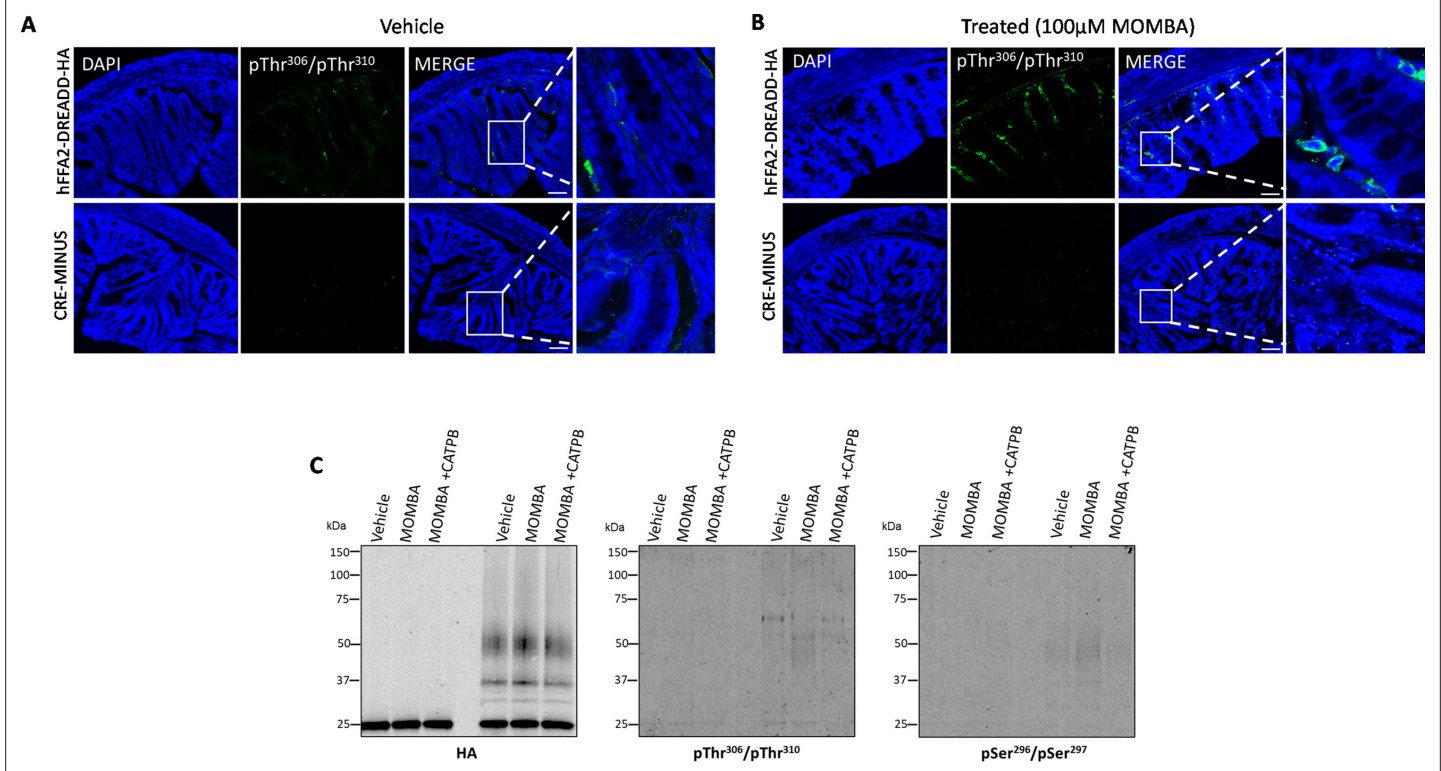

**Figure 6.** 4-Methoxy-3-methyl-benzoic acid (MOMBA) promotes limited phosphorylation of both Ser$^{296}$/Ser$^{297}$ and Thr$^{306}$/Thr$^{310}$ in hFFAR2-DREADD-HA in lower gut enteroendocrine cells. (**A, B**) Colonic tissue isolated from hFFAR2-DREADD-HA (top panels) or CRE-MINUS (bottom panels) mice treated with either vehicle (**A**) or 100 μM MOMBA (**B**). Following fixation tissue sections were immunostained with pThr$^{306}$/pThr$^{310}$ and counterstained with DAPI (scale bar = 100 μm). In the merged images, the box is expanded in the right-hand panels. (**C**) Lysates prepared from tissue samples treated as noted were analysed by probing immunoblots with anti-HA, anti-pThr$^{306}$/pThr$^{310}$, or anti-pSer$^{296}$/pSer$^{297}$. Representative examples are shown.

The online version of this article includes the following source data for figure 6:

**Source data 1.** hFFAR2-DREADD-HA expression in colonic epithelium.

**Source data 2.** pThr$^{306}$/pThr$^{310}$ detects phosphorylation in colonic crypts.

**Source data 3.** pSer$^{296}$/pSer$^{297}$ detects phosphorylation in colonic crypts.

observed in tissue from CRE-MINUS mice, and a further increase in staining following pre-treatment with MOMBA was evident (*Figure 5F*). Higher-level magnification allowed detailed mapping of the location of pThr$^{306}$/pThr$^{310}$ hFFAR2-DREADD-HA (*Figure 5—figure supplement 1*).

Thus, although basal and agonist-regulated phosphorylation of Ser$^{296}$/Ser$^{297}$ hFFAR2-DREADD-HA was evident in both white adipocytes and Peyer's patch immune cells, activation-induced Thr$^{306}$/Thr$^{310}$ phosphorylation is observed in immune cells but not in white adipose tissue.

## Lower gut enteroendocrine cells

We have previously used anti-HA and both anti-GLP-1 and anti-PYY antisera to illustrate the expression of hFFAR2-DREADD-HA in GLP-1 and PYY-positive enteroendocrine cells of the colon of these mice and shown that activation of this receptor enhances release of both GLP-1 (***Bolognini et al., 2019***) and PYY (***Barki et al., 2022***). To extend this we added vehicle or MOMBA to preparations of colonic epithelia from these mice. Subsequent immunostaining with the anti-pThr$^{306}$/pThr$^{310}$ antiserum identified a limited number of widely dispersed cells in MOMBA-treated tissue but minimal staining of the vehicle-treated controls (*Figure 6A and B*). This highlighted groups of spatially scattered cells in which hFFAR2-DREADD-HA became phosphorylated on these residues in an agonist-dependent manner (*Figure 6B*). As an additional specificity control similar experiments were performed with the anti-pThr$^{306}$/pThr$^{310}$ antiserum on tissue from the CRE-MINUS mice. These failed to identify equivalent cells (*Figure 6A and B*). Despite the limited cell expression pattern of anti-HA detected previously in colonic tissue (***Bolognini et al., 2019***), we were again able to specifically capture hFFAR2-DREADD-HA

via HA pull-down (*Figure 6C*). Here, although challenging to detect, we were able to record specific agonist-mediated immuno-recognition of polypeptides of the appropriate molecular mass with both the anti-pThr[306]/pThr[310] and anti-pSer[296]/pSer[297] antisera that were again absent following HA pull-downs conducted in tissue from CRE-MINUS mice (*Figure 6C*). However, compared to the intensity of immunodetection of either anti-pThr[306]/pThr[310] or anti-pSer[296]/pSer[297] in white adipose tissue or Peyer's patch immune cells compared to the extent of anti-HA pull-down, such detection was modest and potentially indicted only limited phosphorylation of these sites in such colonic epithelial cells, even after exposure to MOMBA (*Figure 6C*).

## Wild-type hFFAR2 shows marked similarities in regulated phosphorylation to hFFAR2-DREADD

Whilst the DREADD receptor strategy offers unique control over ligand activation of a GPCR and limits potential activation in tissues by circulating endogenous ligands, it is important to assess if similar outcomes are obtained when using the corresponding wild-type receptor. To do so we next employed Flp-In T-REx 293 cells that allowed doxycycline-induced expression of wild-type hFFAR2-eYFP. As the wild-type and DREADD forms of hFFAR2 are identical within their C-terminal regions it was anticipated that the anti-pSer[296]/pSer[297] and anti-pThr[306]/pThr[310] antisera would be able to detect phosphorylation of these residues in wild-type hFFAR2-eYFP, as noted for hFFAR2-DREADD-eYFP, but instead regulated by SCFAs such as propionate (C3) rather than by MOMBA. Immunoblots using lysates from these cells with either of these antisera showed predominant identification of a group of polypeptides migrating with apparent mass in the region of 60 kDa after treatment of cells induced to express hFFAR2-eYFP with C3 (2 mM, 5 min) (*Figure 7A*). Recognition in this setting of hFFAR2-eYFP by anti-pThr[306]/pThr[310] was almost completely dependent on addition of C3 whilst, although also markedly enhanced by addition of C3, there was some level of identification of hFFAR2-eYFP by anti-pSer[296]/pSer[297] without addition of C3 (*Figure 7A*). Parallel immunoblotting with anti-GFP confirmed the presence of similar levels of hFFAR2-eYFP both with and without treatment with C3 (*Figure 7A*) and confirmed in addition that expression of the receptor construct was lacking if cells harbouring hFFAR2-eYFP had not been induced with doxycycline (*Figure 7A*). As with the effect of MOMBA on the hFFAR2-DREADD-eYFP construct, in studies using the anti-pThr[306]/pThr[310] antiserum, C3-induced recognition of the hFFAR2-eYFP receptor was lacking in the co-presence of the hFFAR2 antagonist CATPB (*Figure 7B*). In addition, treatment of C3-exposed samples to LPP before SDS-PAGE also prevented subsequent hFFAR2-eYFP identification by the anti-pThr[306]/pThr[310] antiserum (*Figure 7B*), confirming this to reflect phosphorylation of the target.

## C3-induced phosphorylation of hFFAR2-eYFP is also detected in immunocytochemical studies

As an alternate means to assess the phosphorylation status of hFFAR2-eYFP in Flp-In T-REx 293 cells, we also performed immunocytochemical studies after cells induced to express the receptor construct had been exposed to either C3 (2 mM) or vehicle. Imaging the presence of eYFP confirmed receptor expression in each case (*Figure 7C and D*) and, in addition, lack of the receptor in cells that had not been exposed to doxycycline. The anti-pThr[306]/pThr[310] antiserum was only able to identify hFFAR2-eYFP after exposure to C3 in this context (*Figure 7C*), whereas for this construct the anti-pSer[296]/pSer[297] antiserum identified the receptor in both vehicle and C3-treated cells (*Figure 7D*).

## Studies in tissues from transgenic mice expressing hFFAR2-HA

To expand such comparisons into native tissues we generated a further transgenic knock-in mouse line. Here, we replaced mFFAR2 with hFFAR2-HA, again with expression of the transgene being controlled in a CRE-recombinase-dependent manner. To assess the relative expression of hFFAR2-HA in this line compared to hFFAR2-DREADD-HA in equivalent tissues of the hFFAR2-DREADD-HA knock-in mice, we isolated both white adipocytes and colonic epithelia from homozygous animals of each line and immunoprecipitated the corresponding receptors using anti-HA. Immunoblotting of such HA immunoprecipitates showed similar levels of the corresponding receptor in each and that the molecular mass of hFFAR2-HA was equivalent to hFFAR2-DREADD-HA (*Figure 8—figure supplement 1*).

In immunocytochemical studies using Peyer's patches isolated from the hFFAR2-HA-expressing mice addition of C3 (10 mM, 20 min) produced a marked increase in detection of hFFAR2-HA by the

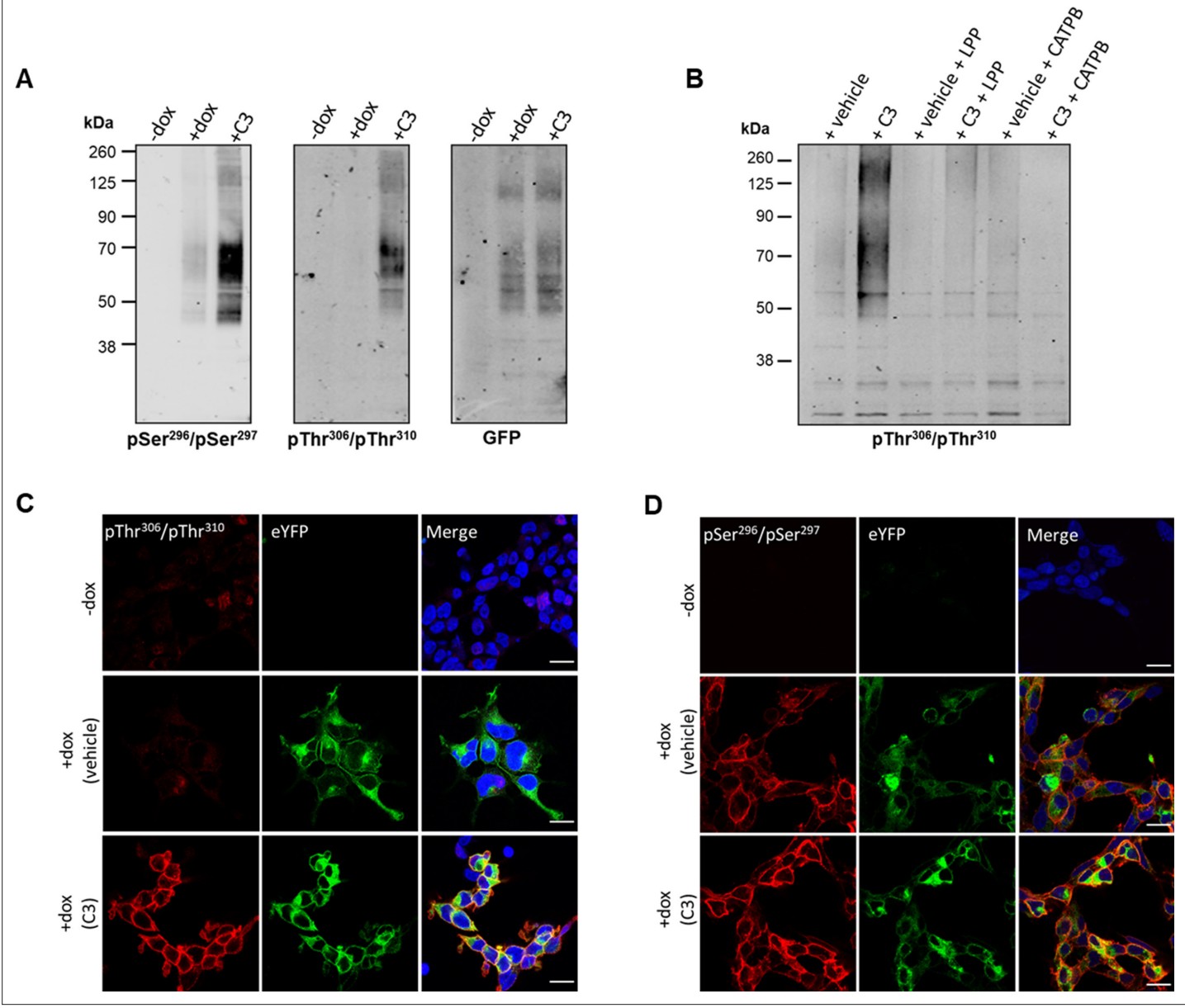

**Figure 7.** Propionate regulates phosphorylation of hFFAR2-eYFP: in vitro studies. Flp-In T-REx 293 cells habouring hFFAR2-eYFP were induced to express the receptor construct (+dox) or not (-dox) and the induced cells were then treated with propionate (C3, 2 mM, 5 min) or vehicle. (**A**) Cell lysates were resolved by SDS-PAGE and then immunoblotted with anti-pSer$^{296}$/pSer$^{297}$ hFFAR2, anti-pThr$^{306}$/pThr$^{310}$ hFFAR2, or anti-GFP. (**B**) Cells induced to express hFFAR2-eYFP were treated with C3 (2 mM, 5 min) or vehicle. Where noted cells were pre-treated with the hFFAR2 antagonist (*S*)-3-(2-(3-chlorophenyl)acetamido)-4-(4-(trifluoromethyl)phenyl)butanoic acid (CATPB) (10 µM, 20 min before agonist addition). Lysates were then prepared and, where indicated, treated with lambda protein phosphatase (LPP). Following SDS-PAGE the samples were immunoblotted with anti-pThr$^{306}$/pThr$^{310}$ hFFAR2. (**C, D**) Cells were doxycycline induced (+dox) or not (-dox) and prepared for immunocytochemistry after treatment with C3 or vehicle and exposed to anti-pThr$^{306}$/pThr$^{310}$ hFFAR2 (**C**) or anti-pSer$^{296}$/pSer$^{297}$ hFFAR2 (**D**) (**red**) whilst direct imaging detected the presence of hFFAR2-eYFP (green). Merged images (right-hand panels) were also stained with DAPI (blue) to identify cell nuclei. Scale bars = 20 µm.

The online version of this article includes the following source data for figure 7:

**Source data 1.** The ability of putative pSer$^{296}$/pSer$^{297}$ antisera to identify phosphorylation of hFFAR2-eYFP.

**Source data 2.** The ability of putative and pThr$^{306}$/Thr$^{310}$ antisera to phosphorylation of hFFAR2-eYFP.

**Source data 3.** GFP control for phosphor-antisera to identify phosphorylation of hFFAR2-eYFP.

**Source data 4.** The ability of pThr$^{306}$/Thr$^{310}$ antisera to detect phosphorylation and inhibition of hFFAR2-eYFP.

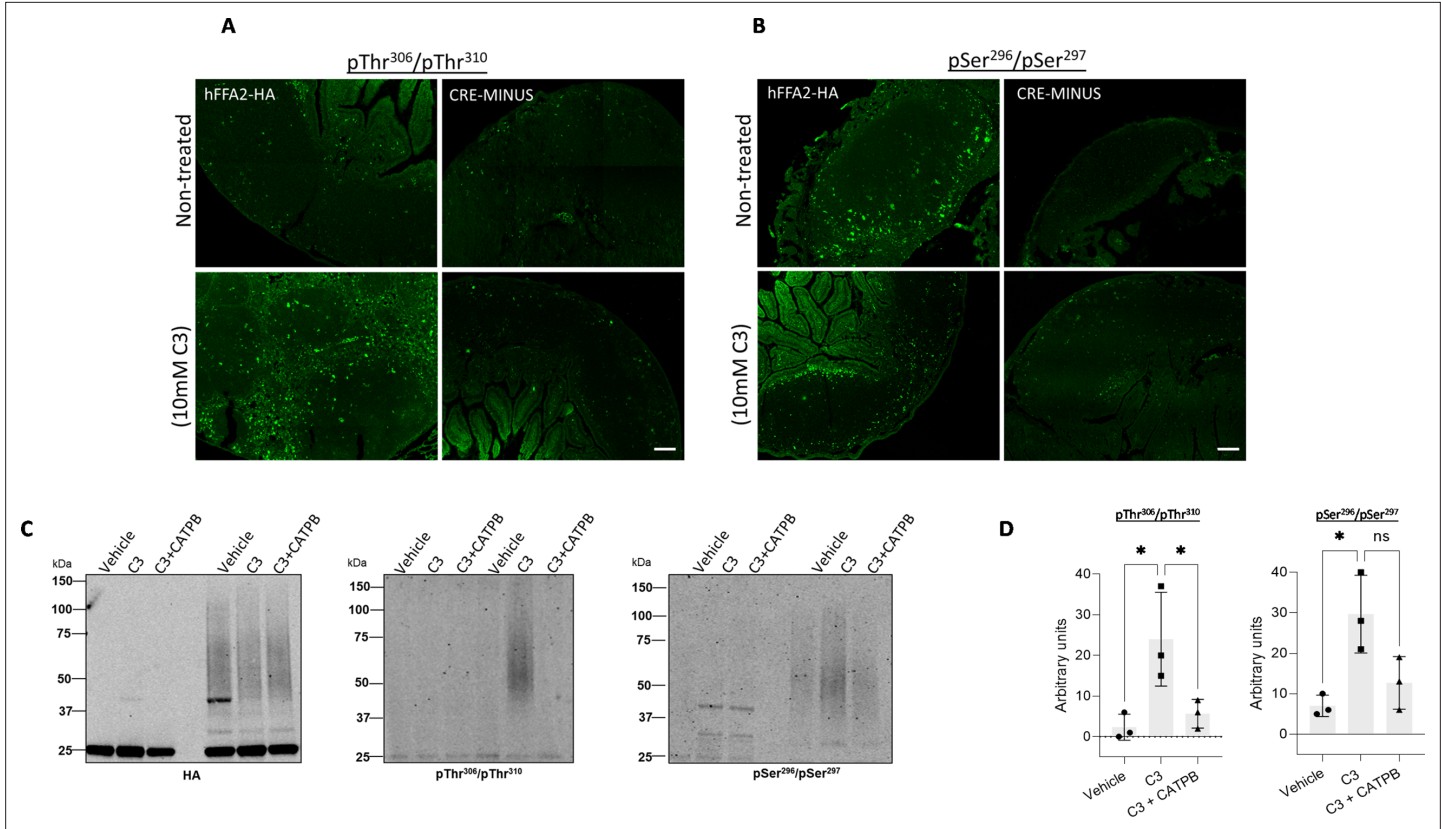

**Figure 8.** C3 induces phosphorylation of both pSer$^{296}$/pSer$^{297}$ and pThr$^{306}$/pThr$^{310}$ in Peyer's patches from hFFAR2-HA-expressing mice. Isolated Peyer's patches and mesenteric lymph nodes from hFFAR2-HA and the corresponding CRE-MINUS mice were exposed to either vehicle or 10 mM C3 for 20 min. Tissue sections were used in immunohistochemical studies, employing either anti-pThr$^{306}$/pThr$^{310}$ (**A**) or anti-pSer$^{296}$/pSer$^{297}$ (**B**) (scale bars = 100 μm). (**C**) Lysates from Peyer's patches isolated from hFFAR2-HA-expressing mice, or the corresponding CRE-MINUS mice, that had been treated with vehicle, C3 (10 mM, 20 min), or C3+(S)-3-(2-(3-chlorophenyl)acetamido)-4-(4-(trifluoromethyl)phenyl)butanoic acid (CATPB) (10 μM, 30 min before agonist) were immunoprecipitated with anti-HA as for the hFFAR2-DREADD-HA-expressing mice in **Figure 5**. Subsequent to SDS-PAGE samples such were probed to detect HA (C, left), anti-pThr$^{306}$/pThr$^{310}$(C, centre), or anti-pSer$^{296}$/pSer$^{297}$(C, right). hFFAR2-HA was detected as a broad smear of protein(s) with M$_r$ centred close to 55 kDa. (**D**) Quantification of pThr$^{306}$/pThr$^{310}$ (left) and pSer$^{296}$/pSer$^{297}$ immunoblots (right) phosphorylation in experiments using tissue from three different mice (means ± SEM), *p<0.05, ns: not significant. Significance was assessed by one-way ANOVA, followed by Tukey's multiple comparisons test (n=3).

The online version of this article includes the following source data and figure supplement(s) for figure 8:

**Source data 1.** hFFAR2-HA expression in Peyer's patches.

**Source data 2.** pThr$^{306}$/pThr$^{310}$ detects agonist-dependent phosphorylation in hFFAR2-HA Peyer's patches.

**Source data 3.** pSer$^{296}$/pSer$^{297}$ detects agonist-dependent phosphorylation in hFFAR2-HA Peyer's patches.

**Source data 4.** Quantification of pSer$^{296}$/pSer$^{297}$ and pThr$^{306}$/pThr$^{310}$ immunoblots for hFFAR2-HA in Peyer's patches.

**Figure supplement 1.** Tissues of transgenic mice express similar levels of hFFAR2-HA and hFFAR2-DREADD-HA.

**Figure supplement 2.** hFFAR2-DREADD-HA is present as multiple differentially N-glycosylated species in adipose tissue, immune, and colonic epithelial cells.

anti-pThr$^{306}$/pThr$^{310}$ antiserum compared to vehicle-treated samples (**Figure 8A**). By contrast, such an effect of C3 was lacking in equivalent tissue isolated from equivalent CRE-MINUS comparator mice (**Figure 8A**). Basal identification of anti-pSer$^{296}$/pSer$^{297}$ staining was significant (**Figure 8B**) and little altered by treatment with C3 (**Figure 8B**). Such basal phosphorylation of Ser$^{296}$/Ser$^{297}$ did appear to be specific however, as this was lacking in tissue from the CRE-MINUS mice (**Figure 8B**). To extend these observations we again used anti-HA to immunoprecipitate hFFAR2-HA from Peyer's patches from hFFAR2-HA-expressing and CRE-MINUS animals after treatment with vehicle, C3, or C3+CATPB. As we observed in samples from hFFAR2-DREADD-HA-expressing mice, subsequent to SDS-PAGE the receptor isolated from such cells migrated as a smear of anti-HA immunoreactivity with apparent

molecular masses ranging from some 50–70 kDa (*Figure 8C*). However, this complex mix clearly corresponded to forms of hFFAR2-HA as such immunoreactivity was once more lacking in HA pull-downs of tissue from CRE-MINUS mice (*Figure 8C*). Immunoblotting such samples with the anti-pThr[306]/pThr[310] antiserum showed clear phosphorylation of pThr[306]/pThr[310] in response to C3 (*Figure 8C*) that increased from essentially undetectable levels in samples of vehicle-treated cells (*Figure 8D*). Like cells from hFFAR2-DREADD-HA-expressing Peyer's patches the effect of C3 was clearly produced directly at the hFFAR2-HA receptor because pre-treatment with CATPB entirely blocked the effect of C3 (*Figure 8C and D*). The anti-pSer[296]/pSer[297] antiserum also showed enhanced detection of hFFAR2-HA in response to treatment with C3 in this setting that was also prevented by pre-treatment with CATPB (*Figure 8C and D*) but this appeared to be less pronounced than in the studies employing anti-pThr[306]/pThr[310] hFFAR2.

## Discussion

The ability to detect activated GPCRs in native tissues can provide insights into how such receptors are regulated and, in drug-discovery and target validation efforts, provide valuable information on target engagement. For most GPCRs, as well as interaction with members of the family of heterotrimeric G proteins, upon agonist occupancy this also results in rapid phosphorylation of various serine and threonine residues, usually within the third intracellular loop and/or the C-terminal tail. Often mediated by members of the GRK family of kinases such phosphorylation allows higher affinity interactions with arrestin proteins. Classically this is linked to desensitisation of receptor-mediated second messenger regulation, as binding of an arrestin usually occludes the G protein binding site. However, now and for a considerable period, it has been established that arrestin interactions can promote distinct signalling functions and that these might be dependent on the detailed architecture of the GPCR-arrestin complex (*Cahill et al., 2017*; *Asher et al., 2022*). This can vary with the pattern of agonist-induced GPCR phosphorylation (*Eiger et al., 2023*; *Butcher et al., 2016*). This concept of phosphorylation barcoding (*Cahill et al., 2017*; *Eiger et al., 2023*; *Butcher et al., 2016*) has been enhanced greatly by both improved approaches to detect such sites via advances in mass spectrometry (*Xiao and Sun, 2018*; *Ives et al., 2022*) and the development and use of phosphorylation-site-specific antisera (*Marsango et al., 2022*; *Butcher et al., 2016*; *Mann et al., 2021*; *Divorty et al., 2022*). Here, guided by combinations of mass spectrometry-identified sites of basal and agonist-regulated phosphorylation of a chemogenetically modified form of the SCFA responsive receptor FFAR2, and informatic predictions of sites that might be or become phosphorylated in an activation-dependent manner, we generated three distinct sets of antisera. For one of these, potentially targeting pSer[324]/pSer[325] hFFAR2-DREADD, we had no direct evidence from mass spectrometry performed on the receptor enriched from an HEK293-derived cell line expressing hFFAR2-DREADD-eYFP of phosphorylation of these sites. Moreover, we failed to find evidence to support such phosphorylation, using the generated antiserum, in either the cell line or the mouse tissues we studied. As we will discuss for the other antisera, this does not however explicitly exclude that these sites might be phosphorylated in other tissues or settings. For the anti-pThr[306]/pThr[310] hFFAR2-DREADD antiserum we also did not find direct mass spectrometry support for phosphorylation of these residues in the HEK293-derived cell line. However, in both immunoblotting studies performed on lysates of these cells and in immunocytochemical studies, this antiserum was able to identify hFFAR2-DREADD-eYFP in a sensitive and agonist activation-dependent manner. Moreover, experiments in which we denuded the receptor of phosphorylation by treatment with LPP demonstrated that the antiserum was only able to identify the phosphorylated receptor and mutation to Ala of both Thr[306] and Thr[310] also eliminated detection of the receptor, confirming the sites of modification.

To assess aspects of potential differential bar-coding we required a second antiserum able to identify a different site(s) of phosphorylation in hFFAR2-DREADD. Here, we did have direct mass spectrometry-derived evidence for basal phosphorylation of Ser[297] and, in addition, agonist-dependent phosphorylation of Ser[296]. In the same way as discussed for the anti-pThr[306]/pThr[310] antiserum we confirmed the specificity of this antiserum and showed in both immunoblotting studies and immunocytochemical studies that there was increased immunoreactivity of this antisera after cells expressing hFFAR2-DREADD were exposed to an appropriate agonist.

Most significantly in the context of phosphorylation bar-coding we were able to show tissue-selective phosphorylation of Thr[306] and Thr[310]. To do so we took advantage of a knock-in transgenic

mouse line that we have characterised extensively (*Bolognini et al., 2019*; *Barki et al., 2022*; *Milligan et al., 2021*). In this line we replaced mouse FFAR2 with hFFAR2-DREADD. In addition, this line has an additional in-frame HA epitope tag added to the C-terminus of the receptor. Both previously, and in the current studies, we have therefore detected the HA tag to identify specific cells expressing the receptor. Herein, however, we built on this to identify the activated receptor and cells directly expressing the activated receptor. These studies provided direct evidence of tissue-specific patterns of agonist-induced FFAR2 receptor bar-coding. In each of the three tissues we explored, white adipose tissue, immune cells in gut Peyer's patches, and in lower intestine enteroendocrine cells, we observed in immunocytochemical studies both basal and agonist-enhanced phosphorylation of Ser$^{296}$/Ser$^{297}$. By contrast, although we observed agonist regulation of Thr$^{306}$/Thr$^{310}$ in both immune cells and enteroendocrine cells, these residues were not phosphorylated in white adipocytes. Hence, we provide clear evidence that the same agonist-GPCR pairing results in a different phosphorylation bar-code in adipocytes compared to Payer's patch immune cells. In addition, the extent of phosphorylation of each of these sites was quantitatively much lower in colonic epithelial than either of the other tissues examined. As we have noted previously (*Fritzwanker et al., 2023*) successful outcomes of these experiments required the maintenance of phosphatase inhibitors throughout.

Importantly we also expanded these studies to a second transgenic knock-in mouse line in which we replaced mFFAR2 with wild-type hFFAR2-HA. This was to assess whether the regulation of receptor phosphorylation we observed in tissues from the hFFAR2-DREADD-HA-expressing line might differ from the wild-type receptor and, if so, whether this could possibly relate to the differences in the way the hFFAR2-DREADD agonist MOMBA activates the receptor compared to the endogenous SCFAs at the wild-type receptor. We considered this unlikely as we have previously shown indistinguishable characteristics of these two ligand-receptor pairs in vitro (*Bolognini et al., 2019*). Importantly, using HA immunoprecipitation from equivalent tissues from these two lines we observed very similar levels of expression of hFFAR2-DREADD-HA and hFFAR2-HA in both white adipose tissue and colonic epithelial preparations and in each the receptor constructs migrated similarly in SDS-PAGE suggesting that co- and post-translational modifications were at least similar. Moreover, as the DREADD variant differs from wild-type receptor only in two amino acids of the orthosteric ligand binding pocket, and the intracellular sections of the two forms are identical (*Bolognini et al., 2019*), we anticipated that the phospho-site-specific antisera would recognise each in an equivalent manner. Overall, comparisons between tissues from the wild-type hFFAR2 and hFFAR2-DREADD-expressing animals were very similar. There was a hint that Ser$^{296}$/Ser$^{297}$ are more highly phosphorylated in the basal state when examining wild-type hFFAR2-HA mice but phosphorylation of Thr$^{306}$/Thr$^{310}$ was entirely agonist-dependent in both Peyer's patches and when using Flp-In T-REx 293 cells.

A further interesting feature, although not inherently linked to bar-coding, is that co-translational N-glycosylation, and potentially other post-translational modifications, of the hFFAR2-DREADD-HA receptor (and probably of hFFAR2-HA) was markedly different in these various tissues. As we were able to use the HA tag to immunoprecipitate the receptor from different tissues, SDS-PAGE and immunoblotting studies showed that the predominant form of the receptor in colonic epithelium migrated as a substantially larger species than that isolated from white adipocytes, and this was also the case when the receptor was immune-enriched from Peyer's patches and associated mesenteric lymph nodes (*Figure 8—figure supplement 2*). De-glycosylation studies using an enzyme able to cleave N-linked carbohydrate confirmed these differences to reflect, at least in part, differential extents of N-glycosylation (*Figure 8—figure supplement 2*).

The basis of the distinct bar-coding induced by MOMBA activation of hFFAR2-DREADD in adipocytes and Peyer's patch immune cells remains to be understood at a molecular level. An obvious possibility is that, at least, the agonist-dependent element of phosphorylation of Ser$^{296}$/Ser$^{297}$ and Thr$^{306}$/Thr$^{310}$ is mediated by distinct GRKs and that the expression patterns or levels of the GRKs vary between these cell types (*Matthees et al., 2021*). These and other possibilities will be assessed in future studies. In arrestin-3 interaction studies performed in HEK293 cells alteration of both Thr$^{306}$/Thr$^{310}$ to Ala resulted in only a modest effect on MOMBA-induced recruitment of the arrestin, whereas alteration of both Ser$^{296}$ and Ser$^{297}$ to Ala produced a more substantial effect. However, it would be inappropriate to attempt to correlate this directly with the phosphorylation observed in different native tissues. FFAR2 is also expressed in other cell types and tissues, including in pancreatic islets. We have also highlighted previously that the G protein selectivity of agonist-activated FFAR2 varies

between different cell types and tissues (*Bolognini et al., 2019*). In time, an extensive analysis across a full range of tissues, which explores G protein, GRK (and potentially other kinases), and arrestin expression, as well as the integration of functions into cell signalling networks and physiological outcomes, will be required to fully appreciate the implications of tissue-selective FFAR2 phosphorylation bar-coding. Despite this, the current studies show clear evidence of how the same ligand-GPCR pairing generates distinct phosphorylation patterns and hence different 'bar-codes' in distinct patho-physiologically relevant tissues.

## Materials and methods

Key reagents were obtained from the following suppliers: VECTASHIELD Antifade Mounting Medium with DAPI (2bscientific, H-1200), anti-HA affinity matrix (Roche, 11815016001), cOmplete ULTRA Tablets, Mini, EASYpack Protease Inhibitor Cocktail (Roche, 5892970001), PhosSTOP (Roche, 4906837001), NuPAGE MOPS SDS Running Buffer (Thermo Fisher Scientific, NP0001), NuPAGE 4–12%, Bis-Tris, 1.0–1.5 mm, Mini Protein Gels (Thermo Fisher Scientific, NP0321BOX), Tris-Glycine Transfer Buffer (Thermo Fisher scientific, LC3675), Laemmli buffer (Sigma, 53401-1VL), Nitrocellulose membrane (Thermo Fisher Scientific, 88018), TSA tetramethylrhodamine (TMR) detection kit (AKOYA Biosciences), SKU NEL702001KT, Tissue culture reagents (Thermo Fisher Scientific). Molecular biology enzymes and the nano-luciferase substrate NanoGlo were from Promega. Polyethylenimine, Linear, MW 25000, Transfection Grade (PEI 25K) obtained from Polysciences. LPP was from New England BioLabs. MOMBA was from Flourochem (018789), whilst CATPB and compound 101 was from Tocris Bioscience.

### Animal maintenance

The generation and characterisation of transgenic hFFAR2-DREADD-HA-expressing and CRE-MINUS mouse lines are described in detail in *Bolognini et al., 2019*. Both male and female mice were used in this study. Breeding, maintenance, and killing of mice conformed to the United Kingdom Home Office regulations (PPL number: PP0894775).

### Mutagenesis of DREADD-FFAR2 receptor-eYFP construct

Site-specific mutagenesis on the hFFAR2 receptor-DREADD-eYFP construct was performed using the Stratagene QuikChange method (Stratagene, Agilent Technologies) and as described in *Marsango et al., 2022*. Primers utilised for mutagenesis were purchased from Thermo Fisher Scientific.

### Cells maintenance, transfection, and generation of cell lines

Cell culture, transfection, and generation of stable cell lines were carried on as described previously (*Sergeev et al., 2017*). HEK293 and Flp-In T-REx 293 cells were authenticated by Northgene (Case Number C-24809a and C-24809b). They were confirmed to be mycoplasma free.

### Cell lysate preparation

Cell lysates were generated from either the various Flp-In T-REx 293 cells following 100 ng/ml doxy-cycline treatment or from HEK293T cells following transient transfection, to express C-terminally eYFP-tagged hFFAR2receptor-DREADD receptor constructs (hFFAR2-DREADDreceptor-eYFP) and prepared as described previously (*Marsango et al., 2022*).

### Cell lysate treatment

To remove phosphate groups, immunocomplexes were treated with LPP at a final concentration of 10 units/µl for 90 min at 30°C before elution with 2× SDS-PAGE sample buffer.

### Receptor immunoprecipitation and immunoblotting assays

eYFP-linked receptor constructs were immunoprecipitated from prepared cell lysates using a GFP-Trap kit (Chromotek) and immunoblotted as described previously (*Marsango et al., 2022*). Membranes were incubated with primary antibodies overnight at 4°C. After overnight incubation the membrane was washed and incubated with secondary antibodies for 2 hr at room temperature.

### Immunocytochemistry

Flp-In T-REx 293 cells expressing hFFAR2-DREADD-eYFP or hFFAR2-eYFP were seeded on poly-D-lysine-coated 13 mm round coverslips in 24-well plates and performed as described previously (*Divorty et al., 2022*).

### Bioluminescence resonance energy transfer-based arrestin-3 recruitment assay

BRET-based arrestin-3 recruitment assay was performed as described in *Marsango et al., 2022*. HEK293T cells were transiently transfected with hFFAR2-DREADD-eYFP (or each of the indicated DREADD-PD mutants) and arrestin-3 fused to nano-luciferase at a ratio of 1:100. Where indicated, cells were pre-treated with 10 µM compound 101 for 30 min at 37°C.

### Tissue dissection

To establish the phosphorylation status of hFFAR2-DREADD-HA or hFFAR2-HA different tissues from hFFAR2-HA, hFFAR2-DREADD-HA, and the corresponding CRE-MINUS mice were dissected. Following cervical dislocation, the entire colon, Peyer's patches, mesenteric lymph nodes, and adipose tissues were quickly removed and placed in ice-cold Krebs-bicarbonate solution (composition in mmol/l: NaCl 118.4, NaHCO$_3$ 24.9, CaCl$_2$ 1.9, Mg$_2$SO$_4$ 1.2, KCl 4.7, KH$_2$PO$_4$ 1.2, glucose 11.7, pH 7.0) supplemented with protease and phosphatase inhibitors. To obtain colonic epithelium preparations, the colon was cut longitudinally and pinned flat on a sylgard-coated Petri dish (serosa down) containing ice-cold sterile oxygenated Krebs solution. The epithelium was gently removed using fine forceps.

### Tissue stimulation

Dissected tissue samples were transferred to warm (30–35°C) Krebs-bicarbonate solution perfused with 95% O$_2$, 5% CO$_2$ and supplemented with protease and phosphatase inhibitors. Tissue samples were challenged with MOMBA (100 µM) (for hFFAR2-DREADD-HA) or C3 (2–10 mM) (for hFFAR2-HA) for 20–30 min. For antagonist treatment, tissue samples were incubated with CATPB (10 µM) for 30 min prior to treatment with MOMBA/C3. Samples intended for western blot analysis were immediately removed and frozen at –80°C. For immunohistochemical analysis tissues were fixed using 4% paraformaldehyde (containing phosphatase inhibitors) for 30 min to 2 hr at room temperature.

### Tissue lysate preparation

Frozen samples were homogenised in RIPA buffer (composition in mmol/l: Tris (base) 50 mM, NaCl 150 mM, sodium deoxycholate 0.5%, Igepal 1%, SDS 0.1%, supplemented with protease phosphatase inhibitor tablets). The samples were further passed through a fine needle (25G) and centrifuged for 15 min at 20,000 × $g$ (4°C). Protein quantification was performed using a BCA protein assay kit (Thermo Fisher Scientific).

### HA-tagged receptor immunoprecipitation and western blot analysis of tissue samples

HA-tagged receptors were immunoprecipitated overnight at 4°C from the tissue lysates using anti-HA Affinity Matrix beads from rat IgG$_1$ (Roche). HA-tagged receptor complexes were centrifuged (2000 × $g$ for 1 min) and washed three times in RIPA buffer. Immune complexes were resuspended in 2× Laemmli sample buffer at 60°C for 5 min. Following centrifugation at 20,000 × $g$ for 2 min, 25 µl of sample was loaded onto an SDS-PAGE on 4–12% Bis-Tris gel. The proteins were separated and transferred onto a nitrocellulose membrane. Non-specific binding was blocked using 5% bovine serum albumin (BSA) in Tris-buffered saline (TBS, 50 mM Tris-Cl, 150 mM NaCl, pH 7.6) supplemented with 0.1% Tween20 for 1 hr at RTP. The membrane was then incubated with appropriate primary antibodies in 5% BSA in TBS-Tween20 overnight at 4°C. Subsequently, the membrane was washed and incubated for 2 hr with secondary antibodies. After washing (3×10 min with TBS-Tween), proteins were visualised using Odyssey imaging system.

### Immunocytochemistry

IHC was performed as previously described by *Bolognini et al., 2019*. Briefly fixed tissues were embedded in paraffin wax and sliced at 5 µM using a microtome. Following deparaffinisation and

antigen retrieval, sections were washed in PBS containing 0.3% Triton X-100. Non-specific binding was blocked by incubating sections for 2 hr at RTP in PBS+0.1% Triton-X+1% BSA+3% goat serum. Subsequently, sections were incubated in appropriate primary antibodies overnight at 4°C. Sections were washed three times in PBS-Triton X-100 and incubated for 2 hr at RTP with species-specific fluorescent secondary antibodies in the dark. Following three washes, sections were mounted with VECTASHIELD Antifade Mounting Medium with DAPI. All images were taken using a Zeiss confocal microscope with Zen software.

## Amplification of HA signal

Signal amplification was performed as previously described by *Barki et al., 2022*. Briefly, sections were incubated with rat anti-HA primary antibody overnight at 4°C, followed by overnight incubation in biotinylated secondary antibody. Immunolabeling was visualised using a TSA TMR detection kit tyramide according to the manufacturer's protocol.

## Antibodies and antisera

The rabbit phospho-site-specific antisera pSer$^{296}$/pSer$^{297}$-hFFAR2 and pThr$^{306}$/pThr$^{310}$-hFFAR2 were developed in collaboration with 7TM Antibodies GmbH.

| Primary antibodies | Source | Catalog no. | WB | ICC/IHC |
|---|---|---|---|---|
| pT$^{306}$/pT$^{310}$-hFFAR2 (phospho-FFAR2) | 7TM | 7TM0226B | 1:1000 | 1:500 |
| pS$^{296}$/pS$^{297}$-hFFAR2 (phospho-FFAR2) | 7TM | 7TM0226A | 1:1000 | 1:500 |
| FFAR2 (non-phospho-FFAR2) | 7TM | 7TM0226N | 1:1000 | 1:500 |
| Anti-HA | Roche | 11867423001 | 1:1000 | 1:250 |
| Anti-GFP | In house | | 1:10,000 | |
| Anti CD11c | ABCAM | | – | 1:500 |
| Anti RORgt | ABCAM | | – | 1:500 |
| Anti CD11 c/b | ABCAM | ab1211 | – | 1:250 |
| Perilipin 1 Monoclonal Antibody (GT2781) | Thermo Fisher Scientific | MA5-27861 | – | 1:250 |
| Secondary antibodies | | | WB | IHC |
| Goat anti-Rat IgG (H+L) Highly Cross-Adsorbed Secondary Antibody, Alexa Fluor Plus 488 | Thermo Fisher Scientific | | – | 1:400 |
| Goat anti-Rabbit IgG (H+L) Highly Cross-Adsorbed Secondary Antibody, Alexa Fluor Plus 488 | Thermo Fisher Scientific | | – | 1:400 |
| Goat anti-Rabbit IgG (H+L) Cross-Adsorbed Secondary Antibody, Alexa Fluor 594 | Thermo Fisher Scientific | | – | 1:400 |
| donkey anti-rabbit IgG Alexa Fluor 647 | ABCAM | | | 1:400 |
| Goat anti Rat IRDYE 800CW | LI-COR Biosciences | 926-32219 | 1:10,000 | – |
| Donkey anti Rabbit IRDYE 800CW | LI-COR Biosciences | 926-32213 | 1:10,000 | – |
| Donkey anti Goat IgG IRDYE 800CW | Thermo Fisher Scientific | | 1:10,000 | |
| Goat anti-Rat IgG (H+L) Secondary Antibody, Biotin (2 ml) | Thermo Fisher Scientific | 31830 | 1:1000 | 1:500 |
| Goat anti-Rabbit IgG (H+L) Secondary Antibody, Biotin | Thermo Fisher Scientific | 65-6140 | 1:1000 | 1:500 |
| Pierce High Sensitivity Streptavidin-HRP | Thermo Fisher Scientific | 21130 | 1:1000 | 1:100 |

## Treatment, membrane preparation, and mass spectrometry analysis of hFFAR2-DREADD

Flp-In T-REx 293 cells harbouring hFFAR2-DREADD-eYFP were cultured to confluence. Receptor expression was induced with 100 ng/ml of doxycycline for 24 hr, followed by 5 min stimulation at

37°C with sorbic acid or MOMBA (100 µM). The cells were washed with ice-cold PBS, and TE buffer containing protease and phosphatase inhibitor cocktails was added to the dishes followed by transfer of cells into pre-chilled tubes. The cells were homogenised and centrifuged at 1500 rpm for 5 min at 4°C. The supernatant was further centrifuged at 50,000 rpm for 45 min at 4°C. The resulting pellet was solubilised in RIPA buffer (50 mM Tris-HCl, 1 mM EDTA, 1 mM EGTA, 1% [vol/vol] Triton X-100, and 0.1% [vol/vol] 2-mercaptoethanol [pH 7.5]) and the protein content was determined using a BCA protein assay kit. hFFAR2-DREADD-eYFP was immunoprecipitated using GFP-Trap Agarose resin (Proteintech) followed by separation on a 4–12% polyacrylamide gel. The gels were stained with Coomassie blue and the bands corresponding to hFFAR2-DREADD-eYFP (~75 kDa) were excised. Gel pieces were destained in 50% EtOH and subjected to reduction with 5 mM DTT for 30 min at 56°C and alkylation with 10 mM iodoacetamide for 30 min before digestion with trypsin at 37°C overnight. The peptides were extracted with 5% formic acid and concentrated to 20 µl. The peptides were subjected to phosphopeptide enrichment using $TiO_2$ beads packed into a StageTip column, and phosphorylated peptides were then eluted, dried, and then injected on an Acclaim PepMap 100 C18 trap and an Acclaim PepMap RSLC C18 column (Thermo Fisher Scientific), using a nanoLC Ultra 2D plus loading pump and nanoLC as-2 autosampler (Eksigent).

Peptides were loaded onto the trap column for 5 min at a flow rate of 5 µl/min of loading buffer (98% water/2% ACN/0.05% trifluoroacetic acid). Trap column was then switched in-line with the analytical column and peptides were eluted with a gradient of increasing ACN, containing 0.1% formic acid at 300 nl/min. The eluate was sprayed into a TripleTOF 5600+electrospray tandem mass spectrometer (AB Sciex Pte. Foster City, USA) and analysed in Information Dependent Acquisition mode, performing 120 ms of MS followed by 80 ms MS/MS analyses on the 20 most intense peaks seen by MS.

The MS/MS data file generated was analysed using the Mascot search algorithm (Matrix Science Inc, Boston, MA, USA), against SwissProt as well as against our in-house database to which we added the protein sequence of interest, using trypsin as the cleavage enzyme. Carbamidomethylation was entered as a fixed modification of cysteine and methionine oxidation, and phosphorylation of serine, threonine, and tyrosine as variable modifications. The peptide mass tolerance was set to 20 ppm and the MS/MS mass tolerance to ±0.1 Da.

Scaffold (version Scaffold_4.8.7, Proteome Software Inc) was used to validate MS/MS-based peptide and protein identifications. Peptide identifications were accepted if they could be established at greater than 20.0% probability. Peptide Probabilities from X! Tandem and Mascot were assigned by the Scaffold Local FDR algorithm. Peptide Probabilities from X! Tandem were assigned by the Peptide Prophet algorithm with Scaffold delta-mass correction. Protein identifications were accepted if they could be established at greater than 95.0% probability and contained at least two identified peptides.

## Data analysis and curve fitting

All data presented represent mean ± SEM of at least three independent experiments. Data analysis and curve fitting were carried out using the GraphPad Prism software package v.10 (GraphPad). Concentration-response data were fit to three-parameter sigmoidal concentration-response curves. In the case of inhibition experiments with antagonists, an equivalent analysis was followed to fit an inverse sigmoidal curve. Statistical analyses were carried out on data derived from at least three independent experiments by performing one-way ANOVA followed by Dunnet's or Tukey's post hoc test as indicated.

## Acknowledgements

These studies were supported by the Biotechnology and Biosciences Research Council UK, grants numbers BB/X001814/1 and BB/S000453/1 (to GM and ABT).

# Additional information

## Funding

| Funder | Grant reference number | Author |
| --- | --- | --- |
| Biotechnology and Biological Sciences Research Council | BB/X001814/1 | Andrew B Tobin Graeme Milligan |
| Biotechnology and Biological Sciences Research Council | BB/S000453/1 | Andrew B Tobin Graeme Milligan |

The funders had no role in study design, data collection and interpretation, or the decision to submit the work for publication.

## Author contributions

Natasja Barki, Formal analysis, Investigation, Writing – review and editing; Laura Jenkins, Sara Marsango, Domonkos Dedeo, Daniele Bolognini, Louis Dwomoh, Aisha M Abdelmalik, Margaret Nilsen, Manon Stoffels, Formal analysis, Investigation; Falko Nagel, Stefan Schulz, Resources, Methodology; Andrew B Tobin, Conceptualization, Funding acquisition, Writing – review and editing; Graeme Milligan, Conceptualization, Supervision, Funding acquisition, Writing - original draft, Project administration, Writing – review and editing

## Author ORCIDs

Natasja Barki  http://orcid.org/0000-0003-4869-4307
Domonkos Dedeo  http://orcid.org/0000-0003-3796-1713
Louis Dwomoh  http://orcid.org/0000-0001-7024-6450
Andrew B Tobin  https://orcid.org/0000-0002-1807-3123
Graeme Milligan  http://orcid.org/0000-0002-6946-3519

## Ethics

The generation and characterization of both transgenic FFA2-DREADD-HA-expressing and CRE-MINUS mouse lines are detailed in Bolognini et al., 2019. All animals were bred as homozygous onto a C57BL/6N background. Male and female mice were used in this study unless otherwise stated. Maintenance and killing of mice followed principles of good laboratory practice in accordance with UK national laws and regulations. All experiments were conducted under a home office licence held by the authors.(PPL number: PP0894775).

Reviewer #1 (Public Review): https://doi.org/10.7554/eLife.91861.3.sa1
Reviewer #2 (Public Review): https://doi.org/10.7554/eLife.91861.3.sa2
Reviewer #3 (Public Review): https://doi.org/10.7554/eLife.91861.3.sa3
Author Response https://doi.org/10.7554/eLife.91861.3.sa4

# Additional files

## Supplementary files
• MDAR checklist

## Data availability

The hFFAR2-DREADD-HA, hFFAR2-HA and corresponding CRE-MINUS mouse lines are available upon request to either ABT or GM. The mass spectrometry proteomics data have been deposited to the ProteomeXchange Consortium via the PRIDE (*Perez-Riverol et al., 2019*) partner repository with the dataset identifier PXD042684. All other data is freely available from GM (Graeme.Milligan@glasgow.ac.uk) or ABT (Andrew.Tobin@glasgow.ac.uk) or through the University of Glasgow's online data repository (DOI: https://doi.org/10.5525/gla.researchdata.1535).

The following datasets were generated:

| Author(s) | Year | Dataset title | Dataset URL | Database and Identifier |
|---|---|---|---|---|
| Tobin AB, Milligan G | 2023 | Phosphorylation bar-coding of Free Fatty Acid receptor 2 is generated in a tissue-specific manner | https://proteomecentral.proteomexchange.org/ui?search=PXD042684 | ProteomeXchange, PXD042684 |
| Tobin AB, Milligan G | 2023 | Phosphorylation bar-coding of Free Fatty Acid receptor 2 is generated in a tissue-specific manner | https://doi.org/10.5525/gla.researchdata.1535 | University of Glasgow, 10.5525/gla.researchdata.1535 |

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
