## [Editor Report · eLife assessment]

In this study, the authors present **important** tools for monitoring distinct tissue-specific patterns of agonist-induced Free Fatty Acid receptor 2 phosphorylation. The work includes several validation experiments, which provide **convincing** evidence that will be beneficial for the scientific community.

---

## [Referee Report · Reviewer #1 (Public Review)]

Summary:

Very systematic generation of phosphosite-specific antisera to monitor FFA2 phosphorylation in native cells and tissues. Provides evidence that FFA2 phosphorylation is tissue-specific.

Strengths:

Technical tour de force, rigorous experimental approaches taking advantage of wt and DREADD versions of FFA2 to make sure that ligand-and receptor-dependent phosphorylations are indeed specific to FFA2.

Weaknesses:

In this reviewer's opinion, the only shortcoming is that the implications of tissue-selective phosphorylation barcoding remain unexplored. However, I understand that tool development is required before tools are used to provide insight into the functional outcomes of receptor regulation by phosphorylation. The study is a technical tour de force to generate highly valuable tools. I have no major criticisms but suggest adding an additional aspect to the discussion as specified below.

Arrestins are highly flexible and dynamic phosphate sensors. If two arrestins have to recognize 800 different phosphorylated GPCRs, is it possible that any barcode serves the same purpose: arrestin recognition followed by signal arrest and internalization? Because phosphorylation barcoding is linked to G protein-independent signaling, which is claimed by some but is experimentally unsupported, and because arrestins don't transduce receptor signals on their own (they only scaffold signaling components and shuttle receptors within cellular compartments), I would also include this option in the discussion, i.e. that the different barcodes are a way nature may have chosen to regulate the location of 800 GPCRs by only 2 arrestins.

---

## [Referee Report · Reviewer #2 (Public Review)]

The strengths of this paper begin with the topic. Specifically, this approaches the question of how GPCR signals are directed to different outcomes under different conditions. There is rich complexity within this question; there are potentially billions of molecules that could interact with >800 human GPCRs and thousands of molecular effectors that may be activated. However, these outcomes are filtered through a small number of GPCR-interacting proteins that direct the signal.

Experimentally, strengths include the initial experimental controls employed in characterizing their ever-important antisera, on which their conclusions hinge. In showing strong agonist-dependent and phosphosite-dependent recognition, as well as the addition of GRK inhibitors and eventually an antagonist and phosphatase treatment, the authors substantiate the role of the antiserum in recognizing their intended motifs. When employed, those antisera overall give clear indications of differences across variables in immunoblots, and while the immunocytochemical studies are qualitative and at times not visually significantly different across all variables, they are in large part congruent with the results of the immunoblots and provide secondary supporting evidence for the author's major claims. One confounding aspect of the immunocytochemical images is the presence of background pThr306/pThr310, like in Figures 4C and 6A and B. In 4A and C, while the immunoblot shows a complete absence of pThr306/pThr310, Figure 4C's immuno image does not. In 6A and B, a similar presence of pThr306/pThr310 is seen in the vehicle image, which is not strikingly over-shown by the MOMBA-treated image. In addition, only Ser/Thr residues of the C-terminus were investigated, while residues of ICL3 have long been known to direct signaling in many GPCRs. Because of the presentations of sequences, it was not clear whether there were residues of ICL3 that have the possibility of being involved.

It may be possible and further testable to show whether the residues that maintain basal phosphorylation could also be tissue-specific, especially considering the presence of pThr306/pThr310 detection in both the Figure 6A immunoblot's vehicle lane (but not MOMBA lane). The aforementioned detection in the immunocytochemical vehicle image could support differential basal phosphorylation in the enteroendocrine cells. Should this be the case, it could have confounded the initial mass-spec screen wherein the Ser residues were basally active in that cell type, while in a distinct cell type that may not be the case. Lastly, should normalized quantification of these images be possible, it may help in clearing up these hard-to-compare visual images.

It is noted that aspects of the writing and presentation may lead to confusion for some readers, but this does not affect the overall significance of the work.

Nevertheless, in terms of the global goal of the authors, the indication of differences in phosphorylation states between tissues is still evident across the experiments. Accordingly, the paper is overall strongly well-researched, well-controlled, and the conclusions made by the authors are data-grounded and not overly extrapolated. Providing direct evidence for the tissue-based branch of the barcode hypothesis is both novel and significant for the field, and the paper leaves room for much more exciting research to be done in the area, opening the door for new questions and hypotheses.

---

## [Referee Report · Reviewer #3 (Public Review)]

Summary:

The authors generate and characterize two phosphospecific antisera for FFA2 receptor and claim a "bar code" difference between white fat and Peyers patches.

Strengths:

The question is interesting and the antibody characterization is convincing.

Weaknesses:

The mass spectrometry analysis is not convincing because the method is not quantitative (no SILAC, TMT, internal standards etc). Figure 1 shows single tryptic peptides with one and two phosphorylation fragmentations as claimed, but there is no data testing the abundance of these so the differences claimed between cell treatment conditions are not established.

The blot analysis cannot distinguish 296/7 but it does convincingly show an agonist increase. Can the authors clarify why the amount of constitutive phosphorylation is much higher in the example blot in Figure 2 than in Figure 3? It would be helpful to quantify this across more than one example, like in Figures 4 and 5 for tissue.

Compound 101 is shown in Figure 2 to block barrestin recruitment. I agree this suggests phosphorylation mediated by GRK2/3 but this is not tested. The new antibodies should be good for this so I don't understand why the indirect approach.

The conditions used to inhibit dephosphorylation are not specified, the method only says "phosphatase inhibitors". How do the authors know that low P at 306/7 in white fat is not a result of dephosphorylation during sample preparation? If these sites are GRK2/3 dependent (see above) then does adipose tissue lack this GRK?

---

## [Author Response]

The following is the authors’ response to the original reviews.

Thank you for the e-mail of 27th September that includes the eLife assessment and reviewers comments on manuscript eLife-RP-RA-2023-91861. We have considered these, added additional data and made various changes to the text as detailed below. We now submit a modified version that we would be happy to view as the ‘Version of Record’.

We are very pleased to note the highly positive reports from the reviewers. The major change we have made is to alter the Introduction to include further consideration of the development of the ‘bar-code’ hypothesis. As highlighted by reviewer 2 the Lefkowitz/Duke University Group have been major proponents of this concept. However, as with many topics their views did not emerge in isolation. Indeed we (specifically Tobin) were developing similar ideas in the same period (see Tobin et al., (2008) Trends Pharmacol Sci 29, 413-420). Moreover, other groups, particularly that of Clark and collaborators at University of Texas, were developing similar ideas using the beta2-adrenoceptor as a model at least as early as this (e.g. Tran et al., (2004) Mol Pharmacol 65, 196-206). As such we have re-written parts of the Introduction to reflect these early studies whilst retaining information on more recent studies that have greatly expanded such early work. This has resulted in the addition of extra references and re-numbering of the Reference section. We have also provided statistical analysis of agonist-induced arrestin interactions with the receptor as requested by a reviewer and performed additional studies to assess the effect of the GRK2/3 inhibitor in agonist-regulation of phosphorylation of the hFFA2-DREADD receptor. This has led to an additional author (Aisha M. Abdelmalik) being added to the paper.

To address first the ‘public reviews’

**Reviewer 1**

1. We agree that we do not at this point explore the implications of the tissue specific barcoding we observe and report. However, as noted by the reviewer these will be studies for the future.

2. The question of why these are only 2 widely expressed arrestins and very many GPCRs is not one we attempt to address here and groups using various arrestin ‘conformation’ sensors are probably much better placed to do so than we are.

**Reviewer 2**

1. It is difficult to address the potential low level of ‘background’ staining in some of the immunocytochemical images versus the ‘cleaner’ background in some of the immunoblotting images. The methods and techniques used are very distinct. However, it should be apparent that the immunoblotting studies are performed (both using cell lines and tissues) post-immunoprecipitation and this is likely to reduce such background to a minimum. This is obviously not the case in the immunocytochemical studies. It is also likely, even though the antisera are immune-selected against the peptide target, there may be some level of immune-recognition this is not limited to the phosphorylated residues.

2. Whilst this reviewer has commented in detail in the ‘recommendations’ section on the use of English, the other reviewers have not, and we do not find the manuscript challenging to follow or read.

**Reviewer 3**

1. We agree that the mass-spectrometry presented is not quantitative. The intention was for the mass spec to be a guide for the development of the antisera used in the study. We have re-written the initial part of the Results section (page 7) to state that phosphorylation of Ser297 was evident in the basal and agonist-stimulated receptor whilst phosphorylation of Ser296 was only evident following agonist addition.

2. Immunoblotting is intrinsically variable as parameters of antiserum titre in re-used samples is not assessed and although we are aware that FFA2 displays a degree of constitutive activity (see for example Hudson et al., (2012) J Biol Chem. 287(49):41195-209) we did not make any specific effort to supress this by, for example, including an inverse agonist ligand. Agonist-regulation of phosphorylation of the receptor, as detected in cell lines by the anti- pThr306/pThr310antiserum, is exceptionally clear cut in all the images displayed, and as we note for the pSer296/pSer297 antiserum this was always, in part, agonist-independent.

The point about compound 101 not being tested directly in the immunoblotting studies performed on the cell line-expressed receptor is a good one. We have now performed such studies which are shown as Figure 2E. These illustrate that the GRK2/3 inhibitor compound 101 does not reduce substantially agonist-induced phosphorylation of the receptor at least as detected by the pThr306/pThr310antiserum or by the pSer296/pSer297 antiserum. Equally this compound had little effect on recognition of the receptor. As the PD2 mutations which correspond to the targets for the pThr306/pThr310antiserum have no significant effect on recruitment of arrestin 3 in response to MOMBA (please see additional statistical analysis in modified Figure 2C) this is perhaps not surprising. Moreover, the PD1 mutations that correspond to the pSer296/pSer297antiserum also, in isolation, only have a partial effect of MOMBA-induced interactions with arrestin 3.

1. The use of phosphatase inhibitors is an integral part of these studies. As noted in Materials we used PhosSTOP (Roche, 4906837001). However, we failed to make it sufficiently clear that this reagent was present throughput sample preparation for both cell lines and tissue studies. This had been specified previously by two of us (SS, FN, see Fritzwanker S, Nagel F, Kliewer A, Stammer V, Schulz S. In situ visualization of opioid and cannabinoid drug effects using phosphosite-specific GPCR antibodies. Commun Biol. 6, 419 (2023)) but we agree this was insufficient and we now correct this oversight by making this explicit in Results.

Recommendations

**Reviewer 1**

Competing interest: We apologise for this typographic error. It is now corrected.

Figures: We have upgraded the figure images to 300dpi and this markedly improves readability

**Reviewer 2**

Revisiting writing: We thank the reviewer for their assessment of the text. However, we do not feel that ‘every sentence in the entire manuscript could be clarified’ is a reasonable statement. Neither of the other reviewers commented on this. Each of the authors read and approved the manuscript.

Figures: see response to Reviewer 1. We have greatly enhanced image quality at this part of the process.

Statistics on Figure 2: We apologise for this oversight. Although there were no significant differences in potency for MOMBA to promote interactions with arrestin-3 to each of the PD mutants versus wild type receptor, there were in terms of maximal effect. Statistical analysis was performed via one-way ANOVA followed by Dunnett’s multiple comparisons test. This is now detailed directly in Figure 2C and its associated legend. As noted by the reviewer there was indeed a highly significant effect of the GRK2/3 inhibitor compound 101 and this is now also noted in Figure 2D and its associated legend.

Units on page 9: pEC50 is considered as Molar by default but we have now specified this.PD1-4: It would be cumbersome to write out (and to read) 8 mutations that make up PD1-4 and hence we think this is specified appropriately in the Figure.

**Reviewer 3**

1. Mass spec: Please see comment point 1 to reviewer 3.

2. Immunoblotting and compound 101: We have done so.

3. Phosphatase inhibition: see public comments, reviewer 3.